# MEASURING UNCERTAINTY CALIBRATION

**Kamil Ciosek    Nicolò Felicioni    Sina Ghiassian**
**Juan Elenter Litwin    Francesco Tonolini    David Gustafsson**
**Eva Garcia-Martin    Carmen Barcena Gonzalez    Raphaëlle Bertrand-Lalo**
Spotify

## ABSTRACT

We make two contributions to the problem of estimating the $L_1$ calibration error of a binary classifier from a finite dataset. First, we provide an upper bound on the calibration error for any classifier where the calibration function has bounded variation. Second, we provide a method of modifying any classifier so that its calibration error can be upper bounded efficiently without significantly impacting classifier performance and without any restrictive assumptions. All our results are non-asymptotic and distribution-free. We conclude by providing advice on how to measure calibration error in practice. Our methods yield practical procedures that can be run on real-world datasets with modest overhead.

## 1  INTRODUCTION

Machine Learning models are increasingly used to drive decision making in many tasks of practical importance. This process is notoriously susceptible (Cohen & Goldszmidt, 2004) to how well model outputs match probabilities of events in the real world, a property known as calibration. An important pre-requisite for ensuring good model calibration is the ability to measure it. This paper focuses precisely on this problem.

The most common approach to calibration is to place model outputs into discrete buckets (Zadrozny & Elkan, 2001; Naeini et al., 2015; Kumar et al., 2019). While this makes the calibration error possible to estimate using elementary tools, it introduces a dilemma. If one treats the estimate of the calibration error obtained by bucketing as a proxy for the calibration error of the classifier before the bucketing, the method is notoriously unreliable, producing different answers depending on the bucketing scheme used (Arrieta-Ibarra et al., 2022). On the other hand, if one treats the bucketing as part of the classifier, classification performance is likely to suffer since the bucketing is unknown to the training process.[1]

Another approach is to frame the problem as a frequentist hypothesis test with the null hypothesis of zero calibration error (Arrieta-Ibarra et al., 2022; Tygert, 2023). This type of method is statistically powerful in detecting deviations from perfect calibration. However, because it focuses on a 'zero-error' null hypothesis, it is primarily designed to decide whether a model is (nearly) perfectly calibrated—not to quantitatively compare degrees of miscalibration between models.[2] Moreover, the proposed cumulative score is not intuitively interpretable and the formal derivation requires the asymptotic regime i.e. a large-enough sample size.[3]

**Contributions**   We make two contributions to the problem of estimating the $L_1$ calibration error of a binary classifier from finite data:

1. **Certified bounds under bounded variation.** We show that if the calibration function has bounded variation, then one can obtain a distribution-free, non-asymptotic *upper bound* on calibration error using a variant of total variation denoising. This provides the first finite-sample guarantees under such a weak structural assumption.

---

[1] For example if the classifier is a neural network, one cannot back-propagate through the bucketed outputs.
[2] One might of course compare test statistics or p-values informally.
[3] Arrieta-Ibarra et al. (2022) do provide an experimental evaluation in the finite-sample regime.

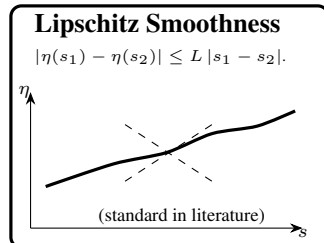 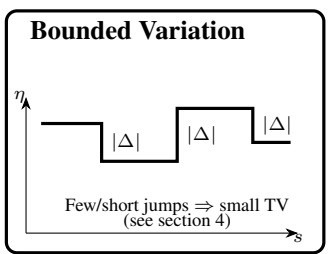 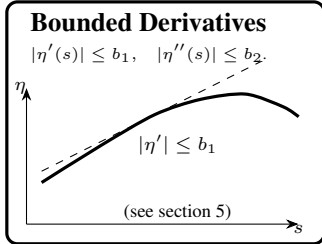

Figure 1: Various assumptions on $\eta(s) = \mathbb{E}[Y \mid S = s]$.

2. **Certified bounds via perturbation to enforce smoothness.** When no assumption on bounded variation is acceptable, we propose a simple perturbation of classifier outputs that guarantees that the calibration function has bounded derivatives. This modification leaves classification performance essentially unchanged, yet enables a kernel-based estimator that yields tighter finite-sample bounds on calibration error.

Both approaches are non-asymptotic and fully distribution-free. We complement the theory with experiments on synthetic and real datasets, and conclude with actionable advice on how to measure calibration error in practice.

## 2 PRELIMINARIES

**Calibration Function** Consider a function $\eta : [0,1] \to [0,1]$. Denote by $p(s)$ the arbitrary distribution of design points (scores). Denote by $p(y|s)$ the Bernoulli distribution with mean $\eta(s)$. We have $\eta(s) = \mathbb{E}[y|s]$, and the function $\eta$ is called the *calibration function*.

**Datasets** Consider an i.i.d dataset of tuples $\{(s_i, y_i)\}_i \sim p(s)p(y|s)$. The dataset is split into two index sets generated independently: training $T$ and validation $V$, where we use the notations $Y_T = \{y_i : i \in T\}$, $Y_V = \{y_i : i \in V\}$, $S_T = \{s_i : i \in T\}$ and $S_V = \{s_i : i \in V\}$. Both these sets are generated independently from any data used to train the classifier.

**Main Objective** The main objective of this paper is to bound the quantity[4]

$$\boxed{\text{CE} \triangleq \mathbb{E}_s\left[|s - \eta(s)|\right],} \tag{1}$$

known (Lee et al., 2023) as the $L_1$ *expected calibration error*, using data from finite (training and validation) datasets. In the next section, we describe the proof approach and structural assumptions on $\eta$ that make this possible.

## 3 STRUCTURE OF RESULTS

**The Need For Assumptions** We begin by noting that estimating the quantity 1 is impossible without making structural assumptions on $\eta$. At a very minimum, for the expectation in equation 1 to be well-defined, one needs measurability. However, measurability alone is not sufficient to ensure the ability to estimate $\eta$ from a finite dataset. In fact, Lee et al. (2023) have shown that even continuity (which implies measurability) is not sufficient, i.e. that there exist choices of $p$ and continuous $\eta$ such that any estimator of the quantity 1 gives an answer wrong by a constant even with infinite data. We therefore need assumptions of a different kind. In this paper, we focus on two alternatives for what they can be.

**Bounded Variation and Bounded Derivatives Assumptions** First, in section 4, we assume that $\eta$ has *bounded variation*, which turns out to be enough to bound the calibration error in a theoretically principled way using a variant of bucketing (with a special process for constructing buckets).

---

[4]We discuss other functionals for measuring calibration and our reasons for picking CE in appendix A.

However, because bounded variation is still a relatively weak assumption, it allows for many nearly pathological functions which the bound has to take into account, adversely impacting sample efficiency. This motivates us to look for stronger assumptions. Specifically, in section 5, we choose to assume $\eta$ has two bounded derivatives. While this assumption ostensibly looks hard to justify, we do in fact show that we can always achieve it by applying a small perturbation to the classifier outputs. We demonstrate in section 7 that this only affects classifier performance in a minimal way in practice. Another possible assumption common in literature is Lipschitz continuity (we compare to a Lipschitz bucketing scheme in section 7). All three assumptions are visualized in figure 1.

**Lower vs Upper Bounds**   As a side note we remark that, while the techniques we develop work for both upper and lower bounds (with high probability), we focus the presentation on upper bounds since we typically wish to certify that the calibration error is small, not that it is large. We also note that all our results are fully distribution-free i.e. the distribution of scores $p(s)$ can be discrete or continuous or a mixture, without any assumptions.

**Main Proof Technique**   Our proof technique consists in constructing a surrogate for $\eta$. We then bound the calibration error as a sum of the calibration error wrt. the surrogate and the error of the surrogate construction process. The construction of the surrogate is different depending on the assumption we are willing to make. Specifically, we use total variation denoising in case of the bounded variation assumption (section 4) and a kernel smoother in case of the bounded-derivatives assumption (section 5).

**Concentration**   In our proof, we also need to quantify concentration of the empirical mean of bounded random variables. We use Bernstein's inequality to do that. Because there are several versions in literature, we quote the one we used. This version matches the work of Mnih et al. (2008). For a sum of $n$ i.id, bounded random variables with mean $\mu$ and support on $[0, 1]$, we have $\mathbb{P}\left[\mu \leq \bar{X}_n + \mathrm{BB}(n, \delta, \widehat{\sigma}^2_{X_i})\right] \geq 1 - \delta$, where $\bar{X}_n := \frac{1}{n}\sum_{i=1}^{n} X_i$, $\widehat{\sigma}^2_{X_i} := \frac{1}{n}\sum_{i=1}^{n}\left(X_i - \bar{X}_n\right)^2$ and

$$\mathrm{BB}(n, \delta, \widehat{\sigma}^2_{X_i}) = \sqrt{\frac{2\,\widehat{\sigma}^2_{X_i}\,\ln(3/\delta)}{n}} + \frac{3\,\ln(3/\delta)}{n}.$$

Here, we used the symbol BB to denote the concentration term in the Bernstein bound. We could have equally used Hoeffding inequality here, but Bernstein is slightly sharper.

## 4   MEASURING CALIBRATION UNDER BOUNDED VARIATION

**Bounded Variation**   To formalize our assumption on $\eta$, we will need the concept of a Bounded Variation space (DeVore & Lorentz, 1993). For a function $f$, we can define total variation on the interval $I$ as

$$\mathrm{TV}(f, I) := \sup\left\{\sum_{i=1}^{n-1} |f(s_{i+1}) - f(s_i)|\right\},$$

where the supremum is taken over the set of all finite sequences $s_1, s_2, \ldots, s_n$ contained in the interval $I$. We say that the function $f$ is of bounded variation, written $f \in \mathrm{BV}([0, 1])$ when $\mathrm{TV}(f, [0, 1]) < \infty$. Our theoretical work in this section is based on the assumption that the calibration function $\eta$ satisfies $\mathrm{TV}(\eta, [0, 1]) \leq V$ for some known constant $V$.

**TV Denoising**   Under the bounded variation assumption, we want to reconstruct $\eta$ from noisy observations. To do so, it is useful to consider the TV denoising estimate $\hat{\eta}_T$. It is obtained by solving the optimization problem

$$\hat{\eta}_T = \arg \min_{v \in [0,1]^{|T|}} \frac{1}{2|T|} \|y_T - v\|_2^2 + \lambda \|Dv\|_1 \tag{2}$$

where $y_T$ is a vector of labels in the training set and $\hat{\eta}_T$ is our approximation of $\eta$ on the training set. The matrix $D \in \mathbb{R}^{(|T|-1)\times|T|}$ is defined as:

$$(Dv)_i = v_{i+1} - v_i, \quad i = 1, 2, \ldots, |T| - 1 \tag{3}$$

and $\lambda > 0$ is a regularization parameter. Mammen & Van De Geer (1997) were the first to have shown results about reconstruction error of similar algorithms, i.e. the difference between $\eta_T$ and $\hat{\eta}_T$. For our purposes, we rely on the fact that, if we set $\lambda = \sqrt{\frac{1}{8|T|} \ln \frac{4(|T|-1)}{\delta_1}}$ in the optimization problem (2), then with probability $1 - \delta_1$ we have

$$\frac{1}{|T|} \|\eta_T - \hat{\eta}_T\|_1 \leq \text{TVB}(\delta_1). \tag{4}$$

Here, we denoted by $\eta_T$ the vector containing the values of $\eta$ on the training set. The term $\text{TVB}(\delta_1)$ diminishes if we increase the size of the training set. This result is proven (and a formula for $\text{TVB}(\delta_1)$ is provided) in appendix B as corollary 1, using proof techniques developed by Hütter & Rigollet (2016). Equation 4 is useful because it measures how close the reconstruction $\hat{\eta}$ is to the true $\eta$ on the training set. However, in order to bound the calibration error, we need a similar guarantee on the population level, i.e. for scores sampled from $p(s)$. To do this, we first define $\hat{\eta}$ for arbitrary scores from its values on the training set

$$\hat{\eta}(s) = \{\hat{\eta}_T\}_{i_\star} \quad \text{where} \quad i_\star = \arg\max_i \{s_i : i \in T, s_i \leq s\}.$$

We can then write a bound similar to 4, but in expectation over $s$.

$$\mathbb{E}_s \left[ |\eta(s) - \hat{\eta}(s)| \right] \leq \text{TVB}(\delta_1) + \underbrace{(V + \hat{V})\sqrt{\frac{\log(2/\delta_2)}{2|T|}} + \sqrt{\frac{1}{2|T|} \log \frac{2}{\delta_3}}}_{\text{PTB}(\delta_2, \delta_3)}, \tag{5}$$

where we denoted part of the right-hand side of the bound with the notation $\text{PTB}(\delta_2, \delta_3)$, which stands for population transfer bound. See Corollary 2 in Appendix B for the proof.

**TV Smoothing as Bucketing** The function $\hat{\eta}$ obtained by TV denoising is piecewise constant. Therefore, a legitimate view of the smoothing process is to consider it as a very special example of a bucketing scheme, where the buckets correspond to the constant pieces of $\hat{\eta}$.

**Bounding the Calibration Error** To upper-bound the calibration error, we rely on a simple but powerful idea. First, imagine that the function $\eta$ in the definition of the calibration error (1) is known. In that case, we could rely on the Bernstein bound to compute the expectation, using the fact that the absolute value in the definition of the calibration error is bounded in $[0, 1]$. However, of course in practice $\eta$ is not known directly and is observable only via noisy observations. This motivates us to proceed by first obtaining an approximation $\hat{\eta}$ to $\eta$ by solving the optimization problem (2) on the training set and then apply Bernstein on the validation set. Note that (2) (and hence $\hat{\eta}$) is computable from the dataset. Intuitively, we can then use the approximation $\hat{\eta}$ in place of $\eta$. Of course, this leads to additional approximation error, which has to be added to the upper bound. Fortunately, the approximation error can be bounded by using equation (4). We now proceed to make this formal.

**Proposition 1** (CE Upper Bound under Bounded Variation). *Construct the surrogate calibration function $\hat{\eta}$ by solving the optimization problem* (2). *With probability $1 - \delta$, we have*

$$\text{CE} \leq \frac{1}{|T|} \sum_{i \in V} |s_i - \hat{\eta}(s_i)| + \text{BB}(|V|, \delta_4, \hat{\sigma}^2_{\{|s_i - \hat{\eta}(s_i)|\}_{i \in V}}) + \text{TVB}(\delta_1) + \text{PTB}(\delta_2, \delta_3),$$

*where $\delta_1 + \delta_2 + \delta_3 + \delta_4 = \delta$ and $\hat{\sigma}^2_{\{|s_i - \hat{\eta}(s_i)|\}_{i \in V}})$ is the empirical variance of the term $|s_i - \hat{\eta}(s_i)|$ on the validation set.*

*Proof.* We write

$$\mathbb{E}_s[|s - \eta(s)|] \leq \underbrace{\mathbb{E}_s[|\hat{\eta}(s) - s|]}_{\text{term I}} + \underbrace{\mathbb{E}_s[|\hat{\eta}(s) - \eta(s)|]}_{\text{term II}} \tag{6}$$

and bound term I and II separately. We begin by bounding term I. Since the term under expectation is bounded in $[0, 1]$ and the validation set is independent of the training set used to obtain $\hat{\eta}$, we use Bernstein inequality with probability $\delta_1$. We bound term II using equation 5. The result follows by picking the delta terms so that $\delta_1 + \delta_2 + \delta_3 + \delta_4 = \delta$. $\square$

Proposition 1 is important because it gives us a handle on bounding the calibration error of any classifier with a bounded variation calibration function in a form computable from observable quantities. Note that the term $\hat{V}$ (the total variation of the surrogate) is computable from data (since the TV solution is piecewise constant).

**When is Bounded Variation Reasonable?**    Verifying that a function is of bounded variation is impossible to do from finite samples only, even if the samples had not been noisy. However, the result in this section is still useful. This is because, in scenarios of practical relevance, $\eta$ is not a completely arbitrary function but one that was obtained from training a classifier. Since the classifier is trying to distinguish positive form negative examples, it is not unreasonable to expect that $\eta$ should be monotone increasing, i.e. that higher scores correspond to a higher chance of seeing a positive label. It is well known that all monotone functions mapping to $[0, 1]$ have total variation bounded by 1, i.e. we can use $V = 1$.

## 5    Measuring Calibration Under Bounded Derivatives

In certain scenarios the bounded variation assumption from the previous section is not suitable. This can be for two reasons–either we don't know the total variation bound $V$ or we want to achieve better sample efficiency. In this section, we address this need by providing a method of obtaining guarantees on calibration for a broad class of classifiers. The main idea is to start with a completely arbitrary classifier and modify it in a way which ensures the ability to bound calibration error but does not harm classification performance.

**Ensuring Smoothness By Perturbation**    The modification amounts to perturbing the probability output of the classifier by a small amount, specified by the perturbation bandwidth. This can be done either only at inference time or at both inference time and training time, to ensure better performance. We experimentally evaluate the effect of the perturbation in section 7 and conclude that there is almost no loss of performance relative to an unmodified classifier for realistic choices of the bandwidth. The calibration function of the perturbed classifier can be written as

$$\eta(s) = \left( \int_0^1 \eta_{\text{orig}}(s_{\text{orig}}) \, k(s \mid s_{\text{orig}}) \, dp(s_{\text{orig}}) \right) \left( \int_0^1 k(s \mid s_{\text{orig}}) \, dp(s_{\text{orig}}) \right)^{-1}. \tag{7}$$

See Appendix D for the derivation of this formula. This expression is a kernel-weighted average of the original calibration function $\eta_{\text{orig}}(s_{\text{orig}})$, smoothened with the kernel $k$.

**Kernel Choice**    Denote by $h$ the amount of the perturbation (the bandwidth). Formally, we define the score $s$ output by the perturbed classifier as a sample from the probability distribution with the PDF defined by the equation

$$k(s \mid s_{\text{orig}}) = \frac{1}{Z(s_{\text{orig}}, h)} \, \text{sech} \left( \frac{s_{\text{orig}} - s}{h} \right), \tag{8}$$

where sech is the hyperbolic secant function and the normalizer $Z(s_{\text{orig}}, h)$ is chosen so that the PDF integrates to one over $[0, 1]$. By construction, the perturbed score is in the interval $[0, 1]$. It turns out that using a perturbation of the form given in equation 8 has significant benefits over using a more conventional choice like the truncated Gaussian. We defer a discussion of our motivation for choosing the perturbation to appendix D.2.

**Bounded Derivatives From Perturbation**    Crucially, the following Lemma, proved in appendix D, shows that the calibration function of the modified classifier has two bounded derivatives, regardless of the properties of the calibration function before the perturbation.

**Lemma 1** (Bounded Derivatives From Perturbation). *For any classifier that perturbs its output scores by replacing an output $s_{orig}$ with a perturbed output $s$ sampled according the PDF in equation 8, the calibration function $\eta$ is twice differentiable. Moreover its first derivative is uniformly bounded by $\frac{1}{2h}$ and its second derivative is uniformly bounded by $\frac{3}{2} \frac{1}{h^2}$.*

**Surrogate of Calibration Function via Nadaraya-Watson Smoothing**   Lemma 1 gives us the derivative bounds that allow us to approximate $\eta$ in a principled way, i.e. bound the smoothing error of a kernel smoother applied to the dataset. We now define $\hat{\eta}$ as

$$\hat{\eta}(s') = \sum_{i \in T} w_i(s') y_i, \quad w_i(s') = \frac{k'(s'|s_i)}{\sum_{j \in T} k'(s'|s_j)}.$$

Then we have:

$$\mathbb{E}_{Y_T}\left[|\hat{\eta}(s') - \eta(s')| S_T\right] \le g_T(s'). \tag{9}$$

where $g_T(s')$ is fully computable from data and diminishes for $s'$ in the support of $p(s)$. See Appendix C for proof and the precise formula for $g_T(s')$. Unlike the choice of the perturbation kernel $k$, the choice of the smoothing kernel $k'$ used to construct the surrogate is an implementation detail. We use Epanechnikov weights with nearest-neighbor fallback and boundary renormalization, tempered by an exponent $\tau = 1.2$ before re-normalizing; since only the "weights sum to one" property matters. One could also explore alternatives to using a kernel smoother. For example, we can see that Lemma 1 implies that $\eta$ is Lipschitz with constant $\frac{1}{2h}$ and use that to bound the error of a bucketing scheme. We do in fact compare to this approach as part of our experiments.

**Bounding The Calibration Error**   We now use equation 9 to bound the calibration error. Again, we stress that all terms in the bound are possible to estimate from data.

**Proposition 2** (CE Upper Bound under Bounded Derivatives). *Assuming $\eta$ is twice differentiable with bounds on the first and second derivative, we have*

$$\text{CE} \le \frac{1}{|V|} \sum_{i \in V} |\hat{\eta}(s_i) - s_i| + \frac{1}{|V|} \sum_{i \in V} g_T(s_i)\ +$$

$$\text{BB}(|V|, \delta_1, \hat{\sigma}^2_{\{|s_i - \hat{\eta}(s_i)|\}_{i \in V}}) + R\,\text{BB}(|V|, \delta_2, \hat{\sigma}^2_{\{\frac{1}{R} g_T(s_i)\}_{i \in V}})$$

*with probability $1 - \delta$ (picked so that $\delta_1 + \delta_2 = \delta$), with the randomness taken over the validation set, conditional on the training set, for a constant $R$ that depends[5] on the kernel $k'$.*

*Proof.* We begin by introducing conditioning on the training set. Because the quantity $|s - \eta(s)|$ doesn't depend on the training set, we can write.

$$\mathbb{E}_s\left[|s - \eta(s)|\right] = \mathbb{E}_s\left[|s - \eta(s)||S_T, Y_T\right]$$

Introduce the notation

$$h(s') = \mathbb{E}_{Y_T}\left[|\hat{\eta}(s') - \eta(s')| S_T\right] = \mathbb{E}_{Y'_T}\left[|\hat{\eta}(s') - \eta(s')| S_T\right],$$

where the last equality holds because we can always rename a bound variable. We can now apply the triangle inequality to get

$$\mathbb{E}_s\left[|s - \eta(s)||S_T, Y_T\right] \le \mathbb{E}_s\left[|s - \hat{\eta}(s)||S_T, Y_T\right] + \mathbb{E}_s\left[h(s)|S_T, Y_T\right]$$
$$= \mathbb{E}_s\left[|s - \hat{\eta}(s)||S_T, Y_T\right] + \mathbb{E}_s\left[h(s)|S_T\right], \tag{10}$$

where the last equality is because $h(s)$ doesn't depend on $Y_T$. We can now apply the Bernstein inequality twice, once for each term on the right hand side of equation (10). Note that both terms under the expectation are bounded in $[0, 1]$. Note also that the validation set is drawn i.i.d.

For the first term, invoking Bernstein (with half the probability budget) conditionally on $S_T, Y_T$. This gives us

$$\mathbb{E}_s\left[|s - \hat{\eta}(s)||S_T, Y_T\right] \le \frac{1}{|V|} \sum_{i \in V} |\hat{\eta}(s_i) - s_i| + \text{BB}(|V|, \delta_1, \hat{\sigma}^2_{\{|s_i - \hat{\eta}(s_i)|\}_{i \in V}}). \tag{11}$$

For the second term in (10), we proceed analogously as before, this time conditioning on $S_T$ and invoking Lemma 6. In order to apply the Bernstein inequality, we introduce a constant $R$ that ensures that $\frac{1}{R} g_T(s') \in [0, 1]$ for all $s'$. We get

$$\mathbb{E}_{s, Y_T}\left[h(s)|S_T\right] \le \frac{1}{|V|} \sum_{i \in V} g_T(s_i) + R\,\text{BB}(|V|, \delta_2, \hat{\sigma}^2_{\{\frac{1}{R} g_T(s_i)\}_{i \in V}}). \tag{12}$$

We get the result by plugging terms (11) and (12) into (10) and using the union bound. $\square$

---

[5]For the truncated Epanechnikov kernel, we set $R = b_1 h + \frac{1}{2} b_2 h^2 + \frac{1}{2}$ where $b_1$ and $b_2$ are uniform bounds on the first and second derivative of $\eta$.

**Implementation Detail (Cross-Fitting).** The theory in the paper is written for a fixed split of the data indices into training and validation. In practice, we use $K$-fold cross-fitting: randomly partition the samples into disjoint folds, and for each fold fit a surrogate eta on the complement while evaluating only on the validation folds, then we aggregate across folds. This preserves the fixed-split assumption (every validation point is scored by a model that did not train on it) while reducing variance and making full use of the data.[6]

## 6 RELATED WORK

**Bucketing** Variants of bucketing were first introduced in meteorology (Murphy & Winkler, 1977) and have been used to approximate the calibration error ever since the ML community started paying attention to calibration (Zadrozny & Elkan, 2001; Niculescu-Mizil & Caruana, 2005; Naeini et al., 2015; Leathart et al., 2017; Guo et al., 2017; Vaicenavicius et al., 2019; Kumar et al., 2019). Nixon et al. (2019) has criticized vanilla bucketing and proposed alternative binning schemes.

**Lipschitz Assumptions** Vaicenavicius et al. (2019); Dimitriadis et al. (2023); Futami & Fujisawa (2024) used the Lipschitz assumption to get a handle on variants of the calibration error. However, the assumption was not justified. Our paper is more general in two ways: bounded variation (as in section 4) is a far weaker assumption then Lipschitz and our perturbation scheme in section 5 guarantees bounded derivatives (and hence the Lipschitz property) rather than just assuming it. Zhang et al. (2020) assumes Hölder continuity, again without justifying it from first principles.

**Kolmogoroff-Smirnoff and Kuiper** Arrieta-Ibarra et al. (2022); Tygert (2023) propose a scheme for measuring calibration based on cumulative sum metrics. By relating these metrics to random walks, it is possible to derive highly efficient frequentist tests where the null hypothesis represents perfect calibration. Our work is different from these approaches in several ways. First, our measured quantity in equation 1 is more interpretable. Second, unlike them, we do not rely on asymptotics of any kind so our result stays theoretically valid for any sample size. Third, our technique can be used to compare two poorly calibrated models with each other while they only support comparisons against a baseline of perfect calibration. Overall, unlike KS-based metrics, which detect mis-calibration but do not yield interpretable bounds, our approach directly certifies an upper bound on CE.

**Impossibility Results** Gupta et al. (2020) studied under what conditions calibration guarantees are theoretically possible, proving formally that one needs assumptions to get a handle on calibration. However, they emphasized binning. On the other hand, our work uses either bounded derivatives via perturbation or a bounded variation assumption, emphasizing reliable, certified measurement of calibration. Another type of impossibility result was provided by Arrieta-Ibarra et al. (2022), who have argued that binning requires an infinite number of samples per bin to accomplish both asymptotic consistency and asymptotic power. However, these results were obtained without structural assumptions on the calibration function $\eta$. In addition, Lee et al. (2023) has shown that assuming continuity of $\eta$ is not enough, i.e. that there are continuous calibration functions which make calibration error impossible to estimate from finite samples. We circumvent these problems when we introduce our assumptions of bounded variation and continuity with bounded derivatives.

**Empirical Studies** Dieye et al. (2025) have conducted an empirical study of several calibration metrics and concluded that the log-loss (or the Brier score) are better proxies for measuring calibration than a bucketing-based ECE estimate. The methods evaluated by Dieye et al. (2025) were all heuristics (they didn't have non-asymptotic guarantees). Maalej et al. (2025) used ECE-type binning in an empirical study of various calibration techniques. Our work is different in that we provide certified upper bounds.

**Separation Between Train and Validation** Our bounds crucially rely on the separation between the training set (used to learn a surrogate of the calibration function) and the validation set (used to measure the calibration error wrt. the surrogate). On the other hand, Kängsepp et al. (2025) interpret

---

[6]We apply such cross-fitting to all three methods we evaluate.

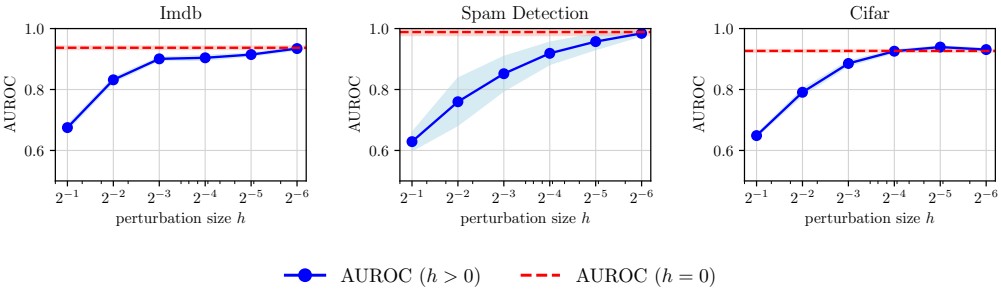

Figure 2: AUROC under Perturbation

the bin-based ECE as a method that performs fitting on the test set. While they provide interesting insights on the ECE, unlike our work, they do not provide certified bounds.

## 7    EXPERIMENTS

**Overview of Experiments**    We perform experiments in two stages. First, we establish that we can afford to perturb classifier outputs by as much as $2^{-6}$ for realistic datasets, without significantly impacting AUROC. Second, we use lemma 1 to certify this guarantees bounded derivatives, allowing us to obtain the calibration error up to about $0.02$ using about $10^7$ samples. While this may seem like a lot of data, we note that estimating the calibration error is an inherently hard problem, which scales poorly with the size of the dataset. In fact, to the best of our knowledge, our estimator is the best currently available approach with certified guarantees for the setting we consider.

**Perturbation vs AUROC**    In figure 2, we measure the effect of perturbing the output probabilities of a classifier on its classification performance, measured by AUROC. We examine three classification tasks: IMDB, Spam Detection, where the classifier is a fine-tuned version of BERT as well as CIFAR where we use a vision transformer. We note that perturbing by $h = 2^{-6}$ costs us almost nothing in terms of AUROC for all three datasets. Note that, to obtain figure 2, the neural network loss was modified to account for the fact that we are applying perturbation at inference time. Details of this are explained in appendix F. The modification is cheap, meaning that the cost of training the network with the modified loss is virtually the same as using the standard cross-entropy loss. The importance of this experiment lies in the fact that it gives a sound methodology for choosing $h$ (we can choose the largest value that does not cause degradation in AUROC).

**Upper Bound Quality vs Sample Size**    The best way to evaluate an estimator is by comparing to the ground truth, which is unavailable for real-world datasets. We therefore introduced four synthetic examples of the calibration functions (see heading of figure 3 for definitions). The figure shows the sample efficiency of bounding calibration error, defined as the gap between the probabilistic upper bound and the ground truth of the calibration error. Since we used synthetic datasets for this experiment, the ground truth is known and we can evaluate the gap exactly. We evaluate four principled methods: our kernel estimator (denoted as NW in the plot), our TV denoising estimator (denoted as TV) and Lipschitz bucketing (denoted as Lip+Bkt in the plot) as well as the ECE heuristic. We observe that all three principled methods are consistent (i.e. the error decreases with increasing dataset size), with NW having the best performance. Interestingly, *the ECE heuristic is extremely competitive for the first three choices of synthetic experiments, but fails completely for the fourth*, in the sense that the error stays large and does not even decrease with increasing dataset size. This behavior is typical of heuristics – they sometimes work well but cannot be counted on for reliable estimation. We believe this underscores the importance of certified bounds.

**Empirical Rates For Principled Estimators**    We will now zoom in onto the quantitative behavior of estimators in figure 3. We can see that the log-log curves are straight lines. We can interpret their slopes as empirical rates. We computed those slopes, obtaining the results shown in Table 1. We can see that the rates obtained in practice very nearly match the theory. We note that, while our

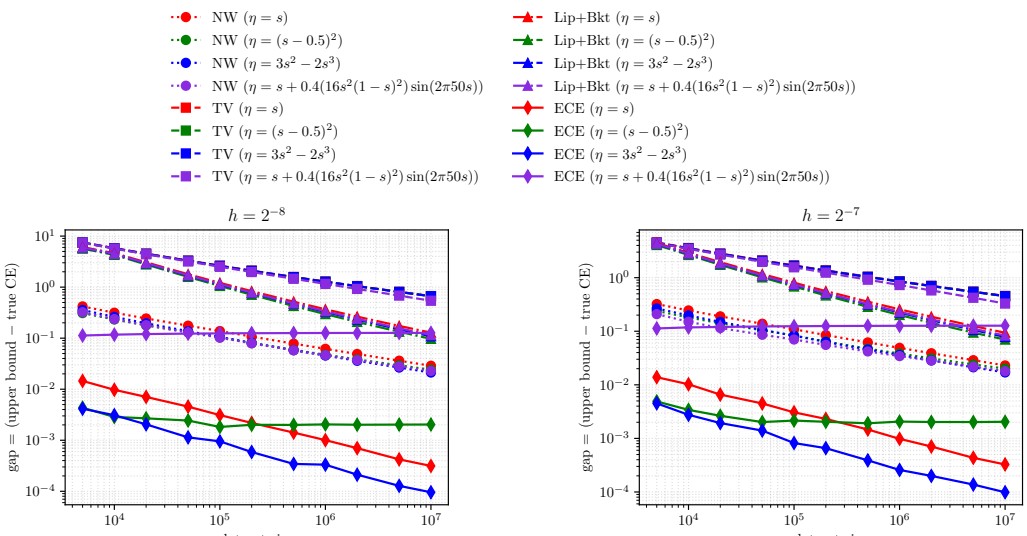

Figure 3: Number of Samples Needed to Achieve Gap (Synthetic Data).

| Method | Empirical Rate | Theoretical Rate | Assumption |
|---|---|---|---|
| NW | $[-0.406, -0.213]$ | $-1/3$ | bounded derivatives |
| TV | $[-0.423, -0.164]$ | $-1/4$ | bounded variation |
| Lip+Bkt | $[-0.574, -0.346]$ | $-1/3$ | Lipschitz smooth |

Table 1: Empirical and theoretical rates for various methods.

NW approach achieves the same rate as Lipschitz bucketing, our constants are better, meaning we produce significantly tighter bounds. Note that TV is still useful in setting where we only want to assume bounded variation.

**Real Data Experiments**   We also wanted to establish how the three bounding techniques work on real data. To do this, we four classification datasets: Amazon Polarity, Civil Comments, Phishing, Yelp Polarity. Since the true calibration error is unknown, unlike the earlier figure, we plot the upper bounds on the calibration error directly (see figure 4). It can be seen that NW smoothing again gives the tightest bounds.

**Note On Repeats and Statistical Significance**   We repeated the all experiments on these plots 64 times and report mean performance. Figure 3 does not contain confidence bars because they are too small to be seen. This makes our results statistically significant.

**Computational Efficiency**   All tested algorithms have at most log-linear time complexity in practice. For kernel smoothing, our implementation uses a sliding window[7], giving linear time complexity. For TV denoising, we used the ProxTV library (Barbero & Sra, 2011; 2018), where the runtime of the 1d denoising variant we use has log-linear complexity. Lipschitz bucketing is linear even with a naive implementation. In practice, it takes about 4 minutes on a single VM to run 64 repeats of all methods for all dataset sizes up to $10^7$, generating a plot such as figure 3.

## 8   LARGE LANGUAGE MODEL USAGE

Large Language Models (ChatGPT 4, ChatGPT 5 and Gemini) were used extensively at all stages in the writing of the paper. Both the theoretical results and the code were verified by the authors. We describe the details of LLM usage in Appendix I.

---

[7]The sliding window method is based on the insight that faraway points have near-zero kernel similarity.

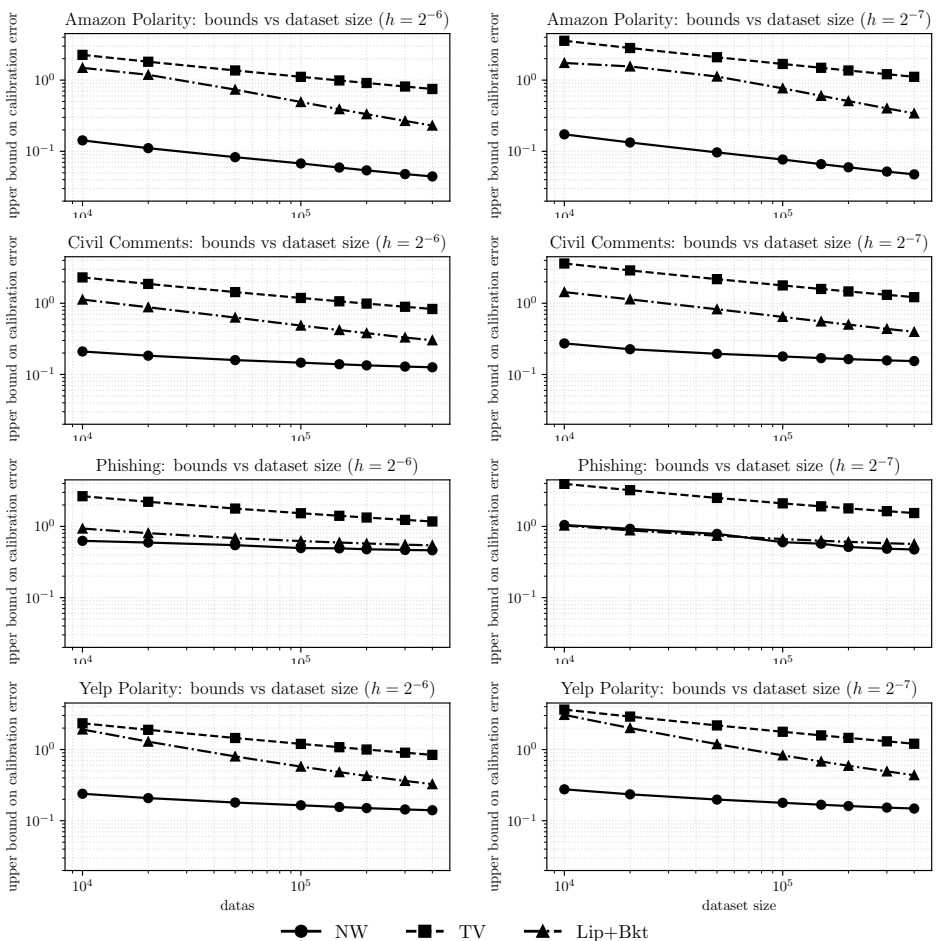

Figure 4: Upper Bounds vs Dataset Size (Real Data).

## 9 PRACTICAL ADVICE

> The preferred technique is to apply a small perturbation and use Proposition 2 (works whether or not training is aware of the perturbation). If perturbations are impossible, assume bounded variation and use Proposition 1 (less sample-efficient). Without either assumption, the problem is intractable in practice.

## 10 CONCLUSIONS

We have proposed two new methods for upper bounding the calibration error of a binary classifier and described their tradeoffs. We empirically demonstrated it is possible to certify an upper bound on the calibration error on a real task of practical importance. We concluded by giving practical advice on how to upper bound calibration error in practice.

## 11 LIMITATIONS

First, while we focused on binary classifiers, we conjecture that our perturbation technique extends naturally to multiclass calibration (future work). Second, while the sample complexity of our NW method is the best we are aware of, it is still the case that one needs about $10^7$ samples to get a certified bound for the L1-calibration error to about $10^{-2}$.

## REPRODUCIBILITY STATEMENT

We provided full proofs for all our claims. Source code is available at https://github.com/spotify-research/calibration.

## ETHICS STATEMENT

Our theoretical results make abstract statements about the quality of estimators. For the experiments, we used readily available public datasets. Neither gives rise to ethics concerns.

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

## A ALTERNATIVE MEASURES OF CALIBRATION ERROR

**Expected $L_q$ norms** The $L_q$ calibration error[8] provides a generalization of the $L_1$ calibration error by measuring deviations between predicted scores and true conditional probabilities using an $L_q$ norm. Formally, it is defined as $\mathbb{E}_s[|\eta(s) - s|^q]^{1/q}$, where $\eta(s)$ denotes the calibration function. By varying $q$, one can adjust the sensitivity to large deviations: higher values of $q$ emphasize larger miscalibration errors, while lower values treat all deviations more uniformly. Among these, the $L_1$ version remains the most interpretable, as it directly quantifies the average absolute discrepancy between predicted probabilities and true event frequencies.

**Expected KL** The expected Kullback–Leibler (KL) divergence, which appears as the calibration term in the Murphy decomposition of the cross entropy scoring rules, measures miscalibration in terms of information loss. It can be viewed as an average over scores of $\text{KL}(\text{Bernoulli}(\eta(s)) \,||\, \text{Bernoulli}(s))$, capturing how much the predictive distribution diverges from the true conditional distribution. This quantity is related to the $L_1$ calibration error through its connection to total variation distance, since total variation provides a first-order (linear) approximation to the KL divergence for small deviations. As with other formulations, the $L_1$ version remains the most interpretable, providing a simple and intuitive measure of average miscalibration.

**Further Justificaition for $L_1$-CE** The $L_1$ calibration error can be justified as a principled measure of calibration through its close connection to the total variation (TV) distance between predictive and true conditional distributions. Specifically, for each score value $s$, the absolute deviation $|\eta(s) - s|$ equals twice the total variation distance between $\text{Bernoulli}(\eta(s))$ and $\text{Bernoulli}(s)$. Since total variation admits the dual characterization $\text{TV}(Q_1, Q_2) = \sup_{|f| \leq 1} |\mathbb{E}_{Q_1}[f] - \mathbb{E}_{Q_2}[f]|$, it bounds the worst-case difference in expectations of any bounded decision function $f$. Consequently, the $L_1$ calibration error measures the expected maximal deviation in decision-relevant quantities that can arise from using the predictive probabilities $s$ instead of the true conditional probabilities $\eta(s)$. In this sense, it has a direct decision-theoretic interpretation: it quantifies how miscalibration can distort the expected utility of any bounded decision rule.

## B TOTAL VARIATION DENOISING

### B.1 PROBLEM FORMULATION

Given a set of training points $s_1, s_2, \ldots, s_{|T|}$, we introduce the vector notations

$$\{\eta_T\}_i = \eta(s_i), \quad \{\hat{\eta}_T\}_i = \hat{\eta}(s_i).$$

---

[8]We use the notation $L_q$ as opposed to the more common $L_p$ because we denoted a probability distribution with the symbol $p$.

We consider a variant of the classical TV denoising problem where we observe binary training data $y_T \in \{0, 1\}^{|T|}$ generated according to:

$$y_{Ti} \sim \text{Bernoulli}(\eta_{Ti}), \quad i = 1, 2, \ldots, |T|. \tag{13}$$

We can always decompose $y_T$ as:

$$y_T = \eta_T + \epsilon_T \tag{14}$$

where $\epsilon_T \in \mathbb{R}^n$ is the noise term representing the deviation of our observations from the expected values.

The TV denoising estimate $\hat{\eta}_T$ is obtained by solving:

$$\hat{\eta}_T = \arg \min_{\eta_T \in [0,1]^{|T|}} \frac{1}{2|T|} \|y_T - \eta_T\|_2^2 + \lambda \|D\eta_T\|_1 \tag{15}$$

where $D \in \mathbb{R}^{(|T|-1) \times |T|}$ is the finite difference operator defined as:

$$(D\eta_T)_i = \eta_{Ti+1} - \eta_{Ti}, \quad i = 1, 2, \ldots, |T| - 1 \tag{16}$$

Since the operator $D$ is linear, it can be represented as a matrix. We abuse notation slightly, also calling this matrix $D$. The variable $\lambda > 0$ in equation 15 is a regularization parameter. We will later derive a way to set $\lambda$ optimally as a function of observable quantities.

We note that by assumption, the total variation of the true parameter vector is bounded: $\|D\eta_T\|_1 \leq V$ for some known constant $V > 0$.

### B.2 AUXILIARY LEMMAS

We first need to establish two useful Lemmas.

**Lemma 2.** *Each row of the matrix $D^{+\top}$ has squared 2-norm at most $\frac{1}{4}|T|$. Here, $D^{+\top}$ is the transpose of the Moore-Penrose pseudo-inverse of the matrix $D$.*

*Proof.* Because $D$ is of full row rank, we have $D^{+\top} = (D^\top(DD^\top)^{-1})^\top = (DD^\top)^{-1}D$. Let us first examine the elements of the matrix $(DD^\top)^{-1}$. The matrix $DD^\top$ is tri-diagonal and its inverse can be computed using the formula derived by Da Fonseca & Petronilho (2001). Using the formula and simplifying gives

$$\{(DD^\top)^{-1}\}_{ij} = \frac{\min(i,j)(|T| - \max(i,j))}{|T|}.$$

Multiplying by $D$ and again simplifying terms, we get

$$\{(D)^{+\top}\}_{jk} = \{(DD^\top)^{-1}D\}_{jk} = \begin{cases} \frac{j}{|T|} & \text{for } j < k \\ \frac{j - |T|}{|T|} & \text{for } j \geq k \end{cases}.$$

The squared-2 norm of row $j$ can be written as:

$$\sum_{k=1}^{j} \left(\frac{j - |T|}{|T|}\right)^2 + \sum_{k=j+1}^{|T|} \left(\frac{j}{|T|}\right)^2 \leq \frac{|T|}{4},$$

where the last inequality is obtained by simplifying and then considering the worst case $j$. $\square$

**Lemma 3.** *Let noise $\epsilon$ be defined as in equation (14). Then we simultaneously have*

$$\|\Pi\epsilon\|_2 \leq t_1, \quad \|D^{+\top}\epsilon\|_\infty \leq t_2$$

*with probability $1 - \delta$, where $t_1$ and $t_2$ are defined as in Lemma 4. The matrix $\Pi$ is defined as $\Pi = \frac{1}{|T|}11^\top$.*

*Proof.* We begin with the bound on $\|\Pi\epsilon\|_2$. First we use the fact that $\Pi = \frac{1}{|T|}11^\top$ to write

$$\|\Pi\epsilon\|_2 = \left\|\frac{1}{|T|}\left(\sum \epsilon_j\right)1\right\|_2 = \frac{1}{|T|}\left|\sum \epsilon_j\right|\|1\|_2 = \frac{1}{\sqrt{|T|}}\left|\sum \epsilon_j\right|.$$

Now, $\epsilon_j$ are centered Bernoulli random variables, hence they are sub-Gaussian with parameter $\frac{1}{4}$. Hence $\sum \epsilon_j$ is sub-Gaussian with parameter $\frac{|T|}{4}$. This allows us to write.

$$P\left(\left|\sum \epsilon_j\right| \geq t\right) \leq 2\exp\left\{-\frac{t^2}{2\frac{|T|}{4}}\right\} = 2\exp\left\{-\frac{2t^2}{|T|}\right\}.$$

Setting the right-hand side to equal $\delta'$, we get $t = \sqrt{\frac{|T|}{2}\ln\frac{2}{\delta'}}$. Hence., with probability $1 - \delta'$, we have

$$\|\Pi\epsilon\|_2 \leq \sqrt{\frac{1}{2}\ln\frac{2}{\delta'}}. \tag{17}$$

We now move to bound the term $\|D^{+\top}\epsilon\|_\infty$. Invoking Lemma 2, we know that every row of $D^{+\top}$ has squared 2-norm upper bounded by $\frac{1}{4}|T|$. Also, like before, each element of $\epsilon$ is sub-Gaussian with parameter $\frac{1}{4}$. Combining these insights, and using the fact that noise terms are independent, the element $j$ of the vector $D^{+\top}\epsilon$ is sub-Gaussian with parameter $\frac{|T|}{4}\frac{1}{4} = \frac{|T|}{16}$. This allows us to write.

$$P\left(\left|\left\{D^{+\top}\epsilon\right\}_j\right| \geq t\right) \leq 2\exp\left\{-\frac{t^2}{2\frac{|T|}{16}}\right\} = 2\exp\left\{-\frac{8t^2}{|T|}\right\}.$$

By the union bound, we have

$$P\left(\left\|D^{+\top}\epsilon\right\|_\infty \geq t\right) = P\left(\max_j\left|\left\{D^{+\top}\epsilon\right\}_j\right| \geq t\right) \leq 2(|T| - 1)\exp\left\{-\frac{8t^2}{|T|}\right\}.$$

Setting the right-hand side to $\delta''$, we obtain $t = \sqrt{\frac{|T|}{8}\ln\frac{2(|T|-1)}{\delta''}}$. This means that with probability $1 - \delta''$, we have

$$\|D^{+\top}\epsilon\|_\infty \leq \sqrt{\frac{|T|}{8}\ln\frac{2(|T|-1)}{\delta''}} \tag{18}$$

The result follows by using (17) and (18) together, setting $\delta' = \delta'' = \delta/2$ and invoking the union bound. □

### B.3 PROOF OF MAIN RESULT ON TV DENOISING

**Lemma 4.** *Define $e = \|\eta_T - \hat{\eta}_T\|_2$. If we set $\lambda = \frac{t_2}{|T|}$ in the optimization problem* (2)*, then with probability $1 - \delta$ we have*

$$\frac{1}{|T|}e^2 \leq \frac{1}{|T|}\left[2t_1^2 + 2t_1\sqrt{t_1^2 + 4t_2V} + 4t_2V\right],$$

*where $t_1 = \sqrt{\frac{1}{2}\ln\frac{4}{\delta}}$ and $t_2 = \sqrt{\frac{|T|}{8}\ln\frac{4(|T|-1)}{\delta}}$.*

The proof follows the technique proposed by Hütter & Rigollet (2016). We make all constants explicit.

*Proof.* Using equation (2) and optimality, we get.

$$\frac{1}{2n}\|y_T - \hat{\eta}_T\|_2^2 + \lambda\|D\hat{\eta}_T\|_1 \leq \frac{1}{2|T|}\|y_T - \eta_T\|_2^2 + \lambda\|D\eta_T\|_1$$

Using the fact that $y_T = \eta_T + \epsilon_T$, we get

$$\frac{1}{2|T|}\|\hat{\eta}_T - \eta_T - \epsilon_T\|_2^2 + \lambda\|D\hat{\eta}_T\|_1 \leq \frac{1}{2|T|}\|\epsilon_T\|_2^2 + \lambda\|D\eta_T\|_1.$$

Expanding the squared term on the left hand side, we get

$$\frac{1}{2|T|}\|\hat{\eta}_T - \eta_T\|_2^2 - \frac{1}{|T|}(\hat{\eta}_T - \eta_T)^\top\epsilon_T + \frac{1}{2|T|}\|\epsilon_T\|_2^2 + \lambda\|D\hat{\eta}_T\|_1 \leq \frac{1}{2|T|}\|\epsilon_T\|_2^2 + \lambda\|D\eta_T\|_1.$$

Canceling the term $\frac{1}{2|T|}\|\epsilon_T\|_2^2$, we get

$$\frac{1}{2|T|}\|\hat{\eta}_T - \eta_T\|_2^2 - \frac{1}{|T|}(\hat{\eta}_T - \eta_T)^\top \epsilon_T + \lambda\|D\hat{\eta}_T\|_1 \le \lambda\|D\eta_T\|_1.$$

Rearranging terms, we get

$$\frac{1}{|T|}\|\hat{\eta}_T - \eta_T\|_2^2 \le \frac{2}{|T|}(\hat{\eta}_T - \eta_T)^\top \epsilon - 2\lambda\|D\hat{\eta}_T\|_1 + 2\lambda\|D\eta_T\|_1. \tag{19}$$

We now focus on the term $(\hat{\eta}_T - \eta_T)^\top \epsilon_T$. Denote by $\Pi$ the projection on the null-space of $D$ (the space spanned by the vector of all ones). Noting that $D^\top D^{+\top} = (I - \Pi)$, we write

$$\begin{aligned}
(\hat{\eta}_T - \eta_T)^\top \epsilon &= (\hat{\eta}_T - \eta_T)^\top (I - \Pi)\epsilon + (\hat{\eta}_T - \eta_T)^\top \Pi\epsilon_T \\
&= (\hat{\eta}_T - \eta_T)^\top D^\top D^{+\top}\epsilon_T + (\hat{\eta}_T - \eta_T)^\top \Pi\epsilon_T \\
&\le \|D(\hat{\eta}_T - \eta_T)\|_1 \|D^{+\top}\epsilon_T\|_\infty + \|(\hat{\eta}_T - \eta_T)\|_2 \|\Pi\epsilon_T\|_2.
\end{aligned}$$

Here, the last inequality follows by using Hölder inequality twice. Invoking Lemma 3, we get

$$(\hat{\eta}_T - \eta_T)^\top \epsilon_T \le \|D(\hat{\eta}_T - \eta_T)\|_1 t_2 + \|(\hat{\eta}_T - \eta_T)\|_2 t_1$$

Plugging back into equation (19), we get

$$\frac{1}{|T|}\|\hat{\eta}_T - \eta_T\|_2^2 \le \frac{2}{|T|}\|D(\hat{\eta}_T - \eta_T)\|_1 t_2 + \frac{2}{|T|}\|(\hat{\eta}_T - \eta_T)\|_2 t_1 - 2\lambda\|D\hat{\eta}_T\|_1 + 2\lambda\|D\eta_T\|_1.$$

Rearranging terms, we get

$$\frac{1}{|T|}\|\hat{\eta}_T - \eta_T\|_2^2 - \frac{2}{|T|}\|(\hat{\eta}_T - \eta_T)\|_2 t_1 \le \frac{2}{|T|}\|D(\hat{\eta}_T - \eta_T)\|_1 t_2 - 2\lambda\|D\hat{\eta}_T\|_1 + 2\lambda\|D\eta_T\|_1.$$

Using the fact that $\|D(\hat{\eta}_T - \eta_T)\|_1 \le \|D\hat{\eta}_T\|_1 + \|D\eta_T\|_1$, we get

$$\frac{1}{|T|}\|\hat{\eta}_T - \eta_T\|_2^2 - \frac{2}{|T|}\|(\hat{\eta}_T - \eta_T)\|_2 t_1 \le \frac{2}{|T|}(\|D\hat{\eta}_T\|_1 + \|D\eta_T\|_1)t_2 - 2\lambda\|D\hat{\eta}_T\|_1 + 2\lambda\|D\eta_T\|_1.$$

Setting $\lambda = \frac{t_2}{|T|}$ and canceling terms, we get

$$\frac{1}{|T|}\|\hat{\eta}_T - \eta_T\|_2^2 - \frac{2}{|T|}\|(\hat{\eta}_T - \eta_T)\|_2 t_1 \le \frac{2}{|T|}\|D\eta_T\|_1 t_2 + 2\frac{t_2}{|T|}\|D\eta_T\|_1 = \frac{4t_2}{|T|}\|D\eta_T\|_1.$$

Using the assumption of bounded total variation, we get $\|D\eta_T\|_1 \le V$, so that

$$\frac{1}{|T|}\|\hat{\eta}_T - \eta_T\|_2^2 - \frac{2}{|T|}\|(\hat{\eta}_T - \eta_T)\|_2 t_1 \le \frac{4t_2}{|T|}V.$$

Denoting $e = \|\hat{\eta}_T - \eta_T\|_2$, we have

$$e^2 - 2et_1 - 4t_2 V \le 0.$$

This implies

$$e \le t_1 + \sqrt{t_1^2 + 4t_2 V}.$$

Squaring and dividing by $|T|$, we get

$$\begin{aligned}
\frac{1}{|T|}e^2 &\le \frac{1}{|T|}\left[t_1^2 + 2t_1\sqrt{t_1^2 + 4t_2 V} + t_1^2 + 4t_2 V\right] \\
&= \frac{1}{|T|}\left[2t_1^2 + 2t_1\sqrt{t_1^2 + 4t_2 V} + 4t_2 V\right].
\end{aligned}$$

$\square$

## B.4 MOVING TO THE L1 NORM

Using Lemma 4 and the Cauchy-Schwarz inequality, we can make the following corollary.

**Corollary 1.** *If we set* $\lambda = \frac{t_2}{|T|}$ *in the optimization problem (2), then with probability* $1 - \delta$ *we have*

$$\frac{1}{|T|}\|\eta_T - \hat{\eta}_T\|_1 \leq \underbrace{\sqrt{\frac{1}{|T|}\left[2t_1^2 + 2t_1\sqrt{t_1^2 + 4t_2V} + 4t_2V\right]}}_{\text{TVB}(\delta)},$$

*where* $t_1 = \sqrt{\frac{1}{2}\ln\frac{4}{\delta}}$, $t_2 = \sqrt{\frac{|T|}{8}\ln\frac{4(|T|-1)}{\delta}}$ *and we have denoted the right-hand-side with* $\text{TVB}(\delta)$.

## B.5 BOUNDED VARIATION POPULATION TRANSFER

One limitation of Corollary 1 is that it only holds on the training set. We now use bounded variation to relate this to a population-based expectation. We start by proving the following Lemma.

**Lemma 5.** *Consider a function* $g : [0, 1] \to [0, 1]$ *of bounded variation* $V_g$. *Consider a sample* $s_1, \ldots, s_n$ *sampled from a distribution with CDF* $F$ *and a empirical CDF* $F_n$, *we have the following with probability* $1 - \delta$.

$$\left|\mathbb{E}_s[g(s)] - \frac{1}{n}\sum_{i=1}^{n}g(s_i)\right| \leq V_g\|F - F_n\|_\infty + \sqrt{\frac{1}{2n}\log\frac{2}{\delta}}$$

*Proof.* Consider the integral

$$\int g dF - \int g dF_n = \int g d(\underbrace{F - F_n}_{H}),$$

where we introduced the notation $H = F - F_n$. Integrating by parts, we obtain

$$\int g dH = -\int H dg + \underbrace{g(1)H(1)}_{0} - g(0)H(0),$$

where the last two terms vanish because $H(1) = 0$. This gives

$$\int g d(F - F_n) = -\int (F - F_n)dg + g(0)H(0)$$

Taking absolute values, we get

$$\left|\int g d(F - F_n)\right| + |g(0)H(0)| \geq \left|\int (F - F_n)dg\right|.$$

Using the fact that $g(0) \in [0, 1]$, we get

$$\left|\int g d(F - F_n)\right| + |H(0)| \geq \left|\int (F - F_n)dg\right|.$$

By the Hoeffding bound, we have with probability $1 - \delta$ that

$$|H(0)| \leq \sqrt{\frac{1}{2n}\log\frac{2}{\delta}}$$

Using the total variation assumption we have,

$$\left|\int g d(F - F_n)\right| \leq V_g\|F - F_n\|_\infty$$

Combining the last three equaitons, we get the desired result. $\square$

Note that the term $\sqrt{\frac{1}{2n}\log\frac{2}{\delta}}$ in lemma 5 is only needed if $F$ assigns nonzero probability mass to the zero score.

We now combine Lemma 5 with corollary 1, obtaining a bound useful for our problem.

**Corollary 2.** *With probability $1 - \delta$, where $\delta_1 + \delta_2 + \delta_3 = \delta$*

$$\mathbb{E}_s\left[|\eta(s) - \hat{\eta}(s)|\right] \leq \underbrace{\frac{1}{|T|}\sum_{i\in T}|\eta(s_i) - \hat{\eta}(s_i)|}_{\leq \text{TVB}(\delta_1)\text{ by Corollary 1}} + (V + \hat{V})\sqrt{\frac{\log(2/\delta_2)}{2|T|}} + \sqrt{\frac{1}{2n}\log\frac{2}{\delta_3}}.$$

*Proof.* Apply Lemma 5 to the function $g(s) = |\eta(s) - \hat{\eta}(s)|$. Apply the DKW theorem to bound the term $\|F - F_n\|_\infty$ with probability $1 - \delta_2$. Apply the property that $V_g \leq V + \hat{V}$ by construction (where $V$ is the total variation of $\eta$ and $\hat{V}$ is the total variation of $\hat{\eta}$). $\qquad\square$

## C  FINITE DERIVATIVE RESULTS

### C.1  BOUNDING THE SMOOTHING ERROR UNDER CONTINUITY

**Lemma 6** (Smoothing Error). *Assume $\eta$ is twice differentiable with the first and second derivatives uniformly bounded by $b_1$ and $b_2$ respectively. Consider a fixed design (set of scores $s_1, \ldots, s_{|T|}$). Denote by $k_{s_i}$ a nonnegative kernel. Denote by $\hat{\eta}$ the Nadaraya-Watson smoothed approximation to $\eta$ obtained from a dataset of $|T|$ samples, defined as*

$$\hat{\eta}(s') = \sum_{i\in T}w_i(s')y_i, \quad w_i(s') = \frac{k_{s_i}(s')}{\sum_{j\in T}k_{s_j}(s')}.$$

*Then we have:*

$$\mathbb{E}_{Y_T}\left[|\hat{\eta}(s') - \eta(s')|\,|S_T\right] \leq g_T(s'),$$

*where*

$$g_T(s') = b_1\sum_{i\in T}w_i(s')|s' - s_i| + b_2\frac{1}{2}\sum_{i\in T}w_i(s')(s' - s_i)^2 + \frac{1}{2}\sqrt{\sum_{i\in T}w_i^2(s')}.$$

*Proof.* We begin by decomposing the term we want to bound.

$$\hat{\eta}(s') - \eta(s') = \underbrace{\left(\sum_{i\in T}w_i(s')\eta(s_i)\right) - \eta(s')}_{\text{bias}} + \underbrace{\sum_{i\in T}w_i(s')(y_i - \eta(s_i))}_{\text{noise}} \tag{20}$$

We will bound the bias and noise term separately. We start with the bias term. Taking the Taylor expansion of $\eta$ around training point $s_i$, evaluated at $s'$, we get

$$\eta(s') = \eta(s_i) + \eta'(s_i)(s' - s_i) + \frac{1}{2}\eta''\xi_i(s')(s' - s_i)^2.$$

for some $\xi_i(s')$. Subtracting $\eta(s_i)$ and summing across the training points, we get

$$\sum_{i\in T}w_i(s')(\eta(s') - \eta(s_i)) = \sum_{i\in T}w_i(s')\eta'(s_i)(s' - s_i) + \frac{1}{2}\sum_{i\in T}w_i(s')\eta''(\xi_i(s'))(s' - s_i)^2.$$

Bounding, we get

$$\left|\sum_{i\in T}w_i(s')(\eta(s') - \eta(s_i))\right| \leq b_1\sum_{i\in T}w_i(s')|s' - s_i| + b_2\frac{1}{2}\sum_{i\in T}w_i(s')(s' - s_i)^2.$$

We now bound the noise term. Applying Jensen's inequality we have

$$\mathbb{E}_{Y_T}\left[\left|\sum_{i \in T} w_i(s')(y_i - \eta(s_i))\right|\Big|S_T\right] \leq \sqrt{\mathbb{E}_{Y_T}\left[\left(\sum_{i \in T} w_i(s')(y_i - \eta(s_i))\right)^2\Big|S_T\right]}.$$

The term $y_i - \eta(s_i)$ is zero mean and sub-Gaussian with variance at most 1/4. Hence we have

$$\sqrt{\mathbb{E}_{Y_T}\left[\left(\sum_{i \in T} w_i(s')(y_i - \eta(s_i))\right)^2\Big|S_T\right]} \leq \frac{1}{2}\sqrt{\sum_{i \in T} w_i^2(s')}.$$

Putting both terms from (20) together, we get the result. $\qquad\square$

## D  SCORE PERTURBATION

### D.1  BOUNDING THE DERIVATIVES OF THE CALIBRATION FUNCTION OF THE MODIFIED CLASSIFIER

We have a dataset of tuples $(s_{\text{orig}}, y)$ where the components are generated as follows.

$$s_{\text{orig}} \sim p(s_{\text{orig}})$$
$$y \sim \text{Bernoulli}(\eta_{\text{orig}}(s_{\text{orig}}))$$

Here, $p(s_{\text{orig}})$ is an arbitrary probability measure for the scores $s_{\text{orig}} \in [0, 1]$ (it can be discrete, continuous, or a mixture). The function $\eta_{\text{orig}}(s_{\text{orig}}) = \mathbb{P}(Y = 1 | S = s_{\text{orig}})$ is the calibration function of the original classifier (before perturbation).

Now, we create a new dataset of tuples $(s_{\text{orig}}, s, y)$ where the components are generated by perturbing the original scores:

$$s_{\text{orig}} \sim p(s_{\text{orig}})$$
$$s \sim k(s \mid s_{\text{orig}})$$
$$y \sim \text{Bernoulli}(\eta_{\text{orig}}(s_{\text{orig}}))$$

Here, for each original score $s_{\text{orig}}$, $k(s \mid s_{\text{orig}})$ is a probability density function (PDF) on $[0, 1]$, which acts as a smoothing kernel. The joint distribution is given by

$$p(s_{\text{orig}}, s, y = 1) = p(s_{\text{orig}}) \underbrace{k(s \mid s_{\text{orig}})}_{p(s|s_{\text{orig}})} \underbrace{\eta_{\text{orig}}(s_{\text{orig}})}_{p(y=1|s_{\text{orig}})}$$

$$p(s_{\text{orig}}, s, y = 0) = p(s_{\text{orig}}) \underbrace{k(s \mid s_{\text{orig}})}_{p(s|s_{\text{orig}})} \underbrace{(1 - \eta_{\text{orig}}(s_{\text{orig}}))}_{p(y=0|s_{\text{orig}})}$$

We are interested in the calibration function of this new perturbed classifier, which is the conditional probability $p(y = 1|s)$. To find it, we first compute the marginals:

$$p(y = 1, s) = \int_0^1 k(s \mid s_{\text{orig}}) \eta_{\text{orig}}(s_{\text{orig}}) \, dp(s_{\text{orig}}) \, ,$$

$$p(y = 0, s) = \int_0^1 k(s \mid s_{\text{orig}})(1 - \eta_{\text{orig}}(s_{\text{orig}})) \, dp(s_{\text{orig}}) \, ,$$

$$p(s) = p(y = 1, s) + p(y = 0, s) = \int_0^1 k(s \mid s_{\text{orig}})(\eta_{\text{orig}}(s_{\text{orig}}) + 1 - \eta_{\text{orig}}(s_{\text{orig}})) \, dp(s_{\text{orig}})$$

$$= \int_0^1 k(s \mid s_{\text{orig}}) \, dp(s_{\text{orig}}) \, .$$

This allows us to compute the calibration function of the perturbed classifier, which we denote by $\eta(s)$:

$$\eta(s) = \frac{p(y = 1, s)}{p(s)} = \frac{\int_0^1 \eta_{\text{orig}}(s_{\text{orig}}) \, k(s \mid s_{\text{orig}}) \, dp(s_{\text{orig}})}{\int_0^1 k(s \mid s_{\text{orig}}) \, dp(s_{\text{orig}})} \, , \ s \in [0, 1]. \tag{21}$$

This expression is a kernel-weighted average of the original calibration function $\eta_{\text{orig}}(s_{\text{orig}})$.

### D.1.1 HYPERBOLIC SECANT KERNEL

We use the hyperbolic secant function as our kernel. For a fixed bandwidth $h > 0$ and for each $s_{\text{orig}} \in [0, 1]$, we define the *sech* kernel $k(s \mid s_{\text{orig}})$:

$$Z(s_{\text{orig}}, h) = \int_0^1 \text{sech}\left(\frac{s - s_{\text{orig}}}{h}\right) ds, \tag{22}$$

$$k(s \mid s_{\text{orig}}) = \frac{1}{Z(s_{\text{orig}}, h)} \text{sech}\left(\frac{s - s_{\text{orig}}}{h}\right) = \frac{1}{Z(s_{\text{orig}}, h)} \frac{2}{e^{\frac{s - s_{\text{orig}}}{h}} + e^{-\frac{s - s_{\text{orig}}}{h}}}. \tag{23}$$

First, we find a closed form equation for $Z(s_{\text{orig}}, h)$ and a lower bound of this quantity, which we will use in later proofs.

**Lemma 7** (Closed form and lower bound for $Z(s_{\text{orig}}, h)$). *For any $h > 0$ and $s_{orig} \in [0, 1]$, the normalization constant in* (22) *admits the closed form*

$$Z(s_{orig}, h) = h\left[\arctan\left(\sinh\left(\frac{1 - s_{orig}}{h}\right)\right) + \arctan\left(\sinh\left(\frac{s_{orig}}{h}\right)\right)\right]. \tag{24}$$

*Moreover, $Z(s_{orig}, h)$ is symmetric about $s_{orig} = \frac{1}{2}$, strictly maximized at $s_{orig} = \frac{1}{2}$, and attains its minimum at the endpoints $s_{orig} \in \{0, 1\}$. In particular,*

$$\min_{s_{orig} \in [0,1]} Z(s_{orig}, h) = Z(0, h) = Z(1, h) = h \arctan\left(\sinh\left(\frac{1}{h}\right)\right). \tag{25}$$

*Proof.* With the substitution $u = (s - s_{\text{orig}})/h$ (so $ds = h\, du$), the limits $s \in [0, 1]$ become $u \in [-s_{\text{orig}}/h, (1 - s_{\text{orig}})/h]$, hence

$$Z(s_{\text{orig}}, h) = \int_0^1 \text{sech}\left(\frac{s - s_{\text{orig}}}{h}\right) ds = h \int_{-s_{\text{orig}}/h}^{(1 - s_{\text{orig}})/h} \text{sech}(u)\, du.$$

Using $\frac{d}{du} \arctan(\sinh u) = \text{sech}\, u$, we obtain

$$Z(s_{\text{orig}}, h) = h\left[\arctan\left(\sinh u\right)\right]_{u = -s_{\text{orig}}/h}^{u = (1 - s_{\text{orig}})/h}$$

$$= h\left(\arctan\left(\sinh\left(\frac{1 - s_{\text{orig}}}{h}\right)\right) + \arctan\left(\sinh\left(\frac{s_{\text{orig}}}{h}\right)\right)\right),$$

which is (24) (we used the oddness of $\sinh$ and $\arctan$).

Now, we will find a lower bound for $Z$. First, differentiate (24) with respect to $s_{\text{orig}}$:

$$\frac{\partial}{\partial s_{\text{orig}}} Z(s_{\text{orig}}, h) = h\left(-\frac{1}{h}\text{sech}\left(\frac{1 - s_{\text{orig}}}{h}\right) + \frac{1}{h}\text{sech}\left(\frac{s_{\text{orig}}}{h}\right)\right) = \text{sech}\left(\frac{s_{\text{orig}}}{h}\right) - \text{sech}\left(\frac{1 - s_{\text{orig}}}{h}\right)$$

Since $x \mapsto \text{sech}\, x$ is strictly decreasing on $[0, \infty)$, we have $\partial_{s_{\text{orig}}} Z(s_{\text{orig}}, h) > 0$ for $s_{\text{orig}} < \frac{1}{2}$, $\partial_{s_{\text{orig}}} Z(s_{\text{orig}}, h) = 0$ at $s_{\text{orig}} = \frac{1}{2}$, and $\partial_{s_{\text{orig}}} Z(s_{\text{orig}}, h) < 0$ for $s_{\text{orig}} > \frac{1}{2}$. Hence $Z$ is strictly increasing on $[0, \frac{1}{2}]$ and strictly decreasing on $[\frac{1}{2}, 1]$, so its maximum is at $s_{\text{orig}} = \frac{1}{2}$ and its minimum is attained at the endpoints $s_{\text{orig}} \in \{0, 1\}$. Evaluating (24) at $s_{\text{orig}} = 0$ or $s_{\text{orig}} = 1$ yields

$$Z(0, h) = Z(1, h) = h \arctan\left(\sinh\left(\frac{1}{h}\right)\right),$$

which is strictly positive for every $h > 0$, proving the uniform lower bound (25). $\square$

Now, we show that $k(s \mid s_{\text{orig}})$ defined in Eq. (23) is a valid probability density function over $[0, 1]$ in the following lemma.

**Lemma 8.** *For any given $s_{orig} \in [0, 1]$ and $h > 0$, the function $k(s \mid s_{orig})$ defined in Eq.* (23) *is a valid probability density function over $[0, 1]$.*

*Proof.* We must verify two conditions: non-negativity and that the integral over $[0, 1]$ is unity.

**(1) Non-negativity.** Recall that for any $t \in \mathbb{R}$, $\mathrm{sech}(t) \in (0, 1]$ is continuous and strictly positive. Therefore the map $s \mapsto \mathrm{sech}\big((s - s_{\mathrm{orig}})/h\big)$ is continuous and strictly positive on $[0, 1]$. Regarding the normalization constant, we showed in lemma 7 that $Z(s_{\mathrm{orig}}, h) > 0$. Since the numerator in $k(s \mid s_{\mathrm{orig}})$ and the denominator $Z(s_{\mathrm{orig}}, h)$ are both positive, we have

$$k(s \mid s_{\mathrm{orig}}) \ = \ \frac{1}{Z(s_{\mathrm{orig}}, h)} \ \mathrm{sech}\Big(\frac{s - s_{\mathrm{orig}}}{h}\Big) \ \geq \ 0 \quad \text{for all } s \in [0, 1].$$

**(2) Unit integral.** Using the definition of $Z(s_{\mathrm{orig}}, h)$,

$$\int_0^1 k(s \mid s_{\mathrm{orig}}) \, ds = \frac{1}{Z(s_{\mathrm{orig}}, h)} \int_0^1 \mathrm{sech}\Big(\frac{s - s_{\mathrm{orig}}}{h}\Big) \, ds = \frac{Z(s_{\mathrm{orig}}, h)}{Z(s_{\mathrm{orig}}, h)} \ = \ 1.$$

Since $k(\cdot \mid s_{\mathrm{orig}}) \geq 0$ on $[0, 1]$ and integrates to 1, it is a valid probability density function on $[0, 1]$.

$\square$

Substituting this specific kernel into our general formula (21) gives the explicit *sech-perturbed calibration*:

$$\eta(s) = \frac{\displaystyle\int_0^1 \eta_{\mathrm{orig}}(s_{\mathrm{orig}}) \, \frac{1}{Z(s_{\mathrm{orig}}, h)} \, \mathrm{sech}\Big(\frac{s - s_{\mathrm{orig}}}{h}\Big) \, dp(s_{\mathrm{orig}})}{\displaystyle\int_0^1 \frac{1}{Z(s_{\mathrm{orig}}, h)} \, \mathrm{sech}\Big(\frac{s - s_{\mathrm{orig}}}{h}\Big) \, dp(s_{\mathrm{orig}})} \ , \ s \in [0, 1]. \tag{26}$$

### D.1.2  A UNIFORM BOUND FOR THE FIRST DERIVATIVE

To analyze the smoothness of $\eta(s)$, we compute its first derivative. Let's define the numerator and denominator of (26) as separate functions:

$$N(s) \ = \ \int_{[0,1]} \eta_{\mathrm{orig}}(s_{\mathrm{orig}}) \, \frac{1}{Z(s_{\mathrm{orig}}, h)} \, \mathrm{sech}\Big(\frac{s - s_{\mathrm{orig}}}{h}\Big) \, dp(s_{\mathrm{orig}}),$$

$$D(s) \ = \ \int_{[0,1]} \frac{1}{Z(s_{\mathrm{orig}}, h)} \, \mathrm{sech}\Big(\frac{s - s_{\mathrm{orig}}}{h}\Big) \, dp(s_{\mathrm{orig}}),$$

so that $\eta(s) = N(s)/D(s)$. To ensure $\eta(s)$ is well-defined, we must show $D(s) \neq 0$. We show this in the following lemma.

**Lemma 9** (Positivity of the denominator). *For the sech kernel, the denominator $D(s)$ is strictly positive for all $s \in [0, 1]$ and any probability measure $p$ on $[0, 1]$.*

*Proof.* Fix $s \in [0, 1]$ and $h > 0$. For every $s_{\mathrm{orig}} \in [0, 1]$ we have $\mathrm{sech}\big((s - s_{\mathrm{orig}})/h\big) > 0$ and $Z(s_{\mathrm{orig}}, h) > 0$ (lemma 7), hence

$$f(s_{\mathrm{orig}}) := \frac{1}{Z(s_{\mathrm{orig}}, h)} \ \mathrm{sech}\Big(\frac{s - s_{\mathrm{orig}}}{h}\Big) > 0 \quad \text{for all } s_{\mathrm{orig}}.$$

Therefore $D(s) = \int_{[0,1]} f(s_{\mathrm{orig}}) \, dp(s_{\mathrm{orig}}) > 0$, since the integral of a strictly positive function with respect to a probability measure is strictly positive. $\square$

Here, we provide the derivative of the sech kernel.

**Lemma 10** (Derivative of the sech kernel). *For every $s_{orig} \in [0, 1]$, the partial derivative of $\mathrm{sech}\Big(\frac{s - s_{orig}}{h}\Big)$ with respect to $s$ is*

$$\frac{\partial}{\partial s} \, \mathrm{sech}\Big(\frac{s - s_{orig}}{h}\Big) \ = \ -\frac{1}{h} \, \mathrm{sech}\Big(\frac{s - s_{orig}}{h}\Big) \tanh\Big(\frac{s - s_{orig}}{h}\Big), \quad \textit{for } s \in [0, 1]. \tag{27}$$

*Proof.* Let $u := (s - s_{\text{orig}})/h$. Since $\operatorname{sech} u = 1/\cosh u$ and $(\cosh u)' = \sinh u$,

$$\frac{d}{du} \operatorname{sech} u = -\frac{\sinh u}{\cosh^2 u} = -\operatorname{sech} u \tanh u.$$

By the chain rule with $\partial u / \partial s = 1/h$,

$$\frac{\partial}{\partial s} \operatorname{sech}\!\left(\frac{s - s_{\text{orig}}}{h}\right) = -\frac{1}{h} \operatorname{sech}\!\left(\frac{s - s_{\text{orig}}}{h}\right) \tanh\!\left(\frac{s - s_{\text{orig}}}{h}\right),$$

which proves (27) for all $s \in [0, 1]$. $\qquad\square$

To compute the derivatives of $N(s)$ and $D(s)$, we need to differentiate under the integral sign. The following lemma provides the justification.

**Lemma 11** (Differentiation under the integral sign). *The derivatives of $N(s)$ and $D(s)$ can be computed as:*

$$N'(s) = \int_{[0,1]} \eta_{orig}(s_{orig}) \frac{1}{Z(s_{orig}, h)} \frac{\partial}{\partial s} \operatorname{sech}\!\left(\frac{s - s_{orig}}{h}\right) dp(s_{orig}), \tag{28}$$

$$D'(s) = \int_{[0,1]} \frac{1}{Z(s_{orig}, h)} \frac{\partial}{\partial s} \operatorname{sech}\!\left(\frac{s - s_{orig}}{h}\right) dp(s_{orig}). \tag{29}$$

*Proof.* Let us start by focusing on $N(s)$. First, we need to find $\frac{\partial}{\partial s} N(s)$. Let's represent $N(s)$ as $N(s) = \int_{[0,1]} f(s_{\text{orig}}, s) dp(s_{\text{orig}})$, where

$$f(s_{\text{orig}}, s) = \eta_{\text{orig}}(s_{\text{orig}}) \frac{1}{Z(s_{\text{orig}}, h)} \operatorname{sech}\!\left(\frac{s - s_{\text{orig}}}{h}\right).$$

Now recall by lemma 10 (using the oddness of tanh),

$$\frac{\partial}{\partial s} \operatorname{sech}\!\left(\frac{s - s_{\text{orig}}}{h}\right) = \frac{1}{h} \operatorname{sech}\!\left(\frac{s - s_{\text{orig}}}{h}\right) \tanh\!\left(\frac{s_{\text{orig}} - s}{h}\right), \quad \text{for } s \in [0, 1].$$

Therefore, we notice that for every $s$ we have

$$\left| \frac{\partial}{\partial s} f(s_{\text{orig}}, s) \right| = \left| \eta_{\text{orig}}(s_{\text{orig}}) \frac{1}{Z(s_{\text{orig}}, h)} \frac{1}{h} \operatorname{sech}\!\left(\frac{s - s_{\text{orig}}}{h}\right) \tanh\!\left(\frac{s_{\text{orig}} - s}{h}\right) \right|$$

$$\leq \frac{1}{h Z(s_{\text{orig}}, h)} =: g(s_{\text{orig}}).$$

Recall that $Z(s_{\text{orig}}, h)$ is strictly positive and lower bounded $Z(s_{\text{orig}}, h) \geq h \arctan\!\left(\sinh(\frac{1}{h})\right)$ from lemma 7. Because of this and the fact that $h > 0$, we have $\sup_{s_{\text{orig}} \in [0,1]} 1/h Z(s_{\text{orig}}, h) < \infty$. Hence $g$ is bounded, so we can apply the dominated convergence theorem (Theorem 11.32 in (Rudin, 1976)), and we may pass the derivative through the integral. An analogous reasoning easily applies to $D(s)$ too. $\qquad\square$

By the quotient rule, the derivative of $\eta(s)$ is:

$$\eta'(s) = \frac{N'(s) D(s) - N(s) D'(s)}{D(s)^2}. \tag{30}$$

**Alternative Representation via Covariance.** The derivative has an elegant alternative representation. Let's define a new probability measure $w_s(ds_{\text{orig}})$ on $[0, 1]$ that depends on $s$. This measure re-weights the original measure $p(s_{\text{orig}})$ based on the kernel:

$$w_s(ds_{\text{orig}}) = \frac{\frac{1}{Z(s_{\text{orig}}, h)} \operatorname{sech}\!\left(\frac{s - s_{\text{orig}}}{h}\right)}{D(s)} dp(s_{\text{orig}}). \tag{31}$$

Note that $\int_0^1 w_s(ds_{\text{orig}}) = \frac{1}{D(s)} \int_0^1 \frac{1}{Z(s_{\text{orig}}, h)} \operatorname{sech}\!\left(\frac{s - s_{\text{orig}}}{h}\right) dp(s_{\text{orig}}) = \frac{D(s)}{D(s)} = 1$.

Now, we use this alternative representation to prove that the derivative of $\eta(s)$ can be expressed as a covariance under $w_s$ in the following proposition.

**Lemma 12.** *The derivative of the perturbed calibration function can be expressed as a covariance under the measure* $w_s(ds_{orig})$*:*

$$\eta'(s) = \frac{1}{h} \ \mathrm{Cov}_{S \sim w_s}\left( \eta_{orig}(S), \ \tanh\left(\frac{S - s}{h}\right) \right) \ , \quad s \in [0, 1].$$

*Proof.* Starting from (30) and writing $T(u) := \tanh\left(\frac{s-u}{h}\right)$, lemma 11 together with lemma 10 gives

$$\frac{N'(s)}{D(s)} = \frac{1}{D(s)} \int_{[0,1]} \eta_{\mathrm{orig}}(s_{\mathrm{orig}}) \frac{1}{Z(s_{\mathrm{orig}}, h)} \frac{\partial}{\partial s} \mathrm{sech}\left(\frac{s - s_{\mathrm{orig}}}{h}\right) dp(s_{\mathrm{orig}})$$

$$= -\frac{1}{h} \ \mathbb{E}_{w_s}\big[\eta_{\mathrm{orig}}(S) \, T(S)\big],$$

and

$$\frac{D'(s)}{D(s)} = \frac{1}{D(s)} \int_{[0,1]} \frac{1}{Z(s_{\mathrm{orig}}, h)} \frac{\partial}{\partial s} \mathrm{sech}\left(\frac{s - s_{\mathrm{orig}}}{h}\right) dp(s_{\mathrm{orig}}) = -\frac{1}{h} \ \mathbb{E}_{w_s}\big[T(S)\big].$$

Since $\eta(s) = \frac{N(s)}{D(s)} = \mathbb{E}_{w_s}[\eta_{\mathrm{orig}}(S)]$, the quotient rule (30) yields

$$\begin{aligned}
\eta'(s) &= \frac{N'(s)}{D(s)} - \frac{N(s)}{D(s)} \frac{D'(s)}{D(s)} \\
&= -\frac{1}{h} \ \mathbb{E}_{w_s}\big[\eta_{\mathrm{orig}}(S) \, T(S)\big] \ + \ \frac{1}{h} \ \mathbb{E}_{w_s}[\eta_{\mathrm{orig}}(S)] \ \mathbb{E}_{w_s}[T(S)] \\
&= -\frac{1}{h} \Big( \mathbb{E}_{w_s}[\eta_{\mathrm{orig}}(S) \, T(S)] - \mathbb{E}_{w_s}[\eta_{\mathrm{orig}}(S)] \ \mathbb{E}_{w_s}[T(S)] \Big) \\
&= -\frac{1}{h} \ \mathrm{Cov}_{S \sim w_s}\left( \eta_{\mathrm{orig}}(S), \ \tanh\left(\frac{s - S}{h}\right) \right).
\end{aligned}$$

Because $\tanh$ is odd, this is equivalently

$$\eta'(s) = \frac{1}{h} \ \mathrm{Cov}_{S \sim w_s}\left( \eta_{\mathrm{orig}}(S), \ \tanh\left(\frac{S - s}{h}\right) \right).$$

$\square$

Using this alternative representation as a covariance, we provide a uniform bound for the first derivative in the following corollary.

**Corollary 3** (Uniform Lipschitz bound). *For the* sech *kernel and any bandwidth* $h > 0$*, the perturbed calibration function* $\eta$ *is Lipschitz continuous with a constant independent of* $p$ *and* $\eta$*:*

$$\sup_{s \in [0,1]} |\eta'(s)| \ \leq \ \frac{\tanh(1/h)}{2h} \ \leq \ \frac{1}{2h}.$$

*Proof.* By lemma 12, $\eta'(s) = \frac{1}{h} \mathrm{Cov}_{w_s}\left( \eta_{\mathrm{orig}}(S), T \right)$ with $T = \tanh\left(\frac{S-s}{h}\right)$. By Cauchy–Schwarz, we know that

$$|\mathrm{Cov}(X, Y)| \leq \sqrt{\mathbb{V}(X) \, \mathbb{V}(Y)},$$

where with $\mathbb{V}(\cdot)$ we indicate the variance of a random variable.

Also, Popoviciu's inequality says that, if $Z \in [a, b]$, then we have a bound for its variance $\mathbb{V}(Z)$:

$$\mathbb{V}(Z) \leq \frac{(b - a)^2}{4}.$$

Since $\eta_{\mathrm{orig}}(S) \in [0, 1]$ and $T \in [-\tanh(1/h), \tanh(1/h)]$, Cauchy–Schwarz plus Popoviciu's inequality gives

$$\frac{1}{h} |\mathrm{Cov}(\eta_{\mathrm{orig}}(S), T)| \ \leq \ \frac{1}{h} \sqrt{\frac{1}{4} \cdot \frac{(2 \tanh(1/h))^2}{4}} = \frac{1}{h} \sqrt{\frac{\tanh^2(1/h)}{4}} = \frac{\tanh(1/h)}{2h}.$$

Noticing that $\tanh(1/h) \leq 1$ yields the claim. $\square$

### D.1.3 SECOND DERIVATIVE AND A UNIFORM BOUND

Let us focus now on bounding the second derivative. First, we will find the second derivative of the kernel in the next lemma.

**Lemma 13** (Second derivative of the kernel). *For every $s_{orig} \in [0, 1]$ and $h > 0$,*

$$\frac{\partial^2}{\partial s^2} \operatorname{sech}\left(\frac{s - s_{orig}}{h}\right) = \frac{1}{h^2} \operatorname{sech}\left(\frac{s - s_{orig}}{h}\right) \left(2 \tanh^2\left(\frac{s - s_{orig}}{h}\right) - 1\right), \quad s \in [0, 1].$$

*Proof.* Let $u = (s - s_{\text{orig}})/h$. Using $\frac{d}{du} \operatorname{sech} u = -\operatorname{sech} u \tanh u$ and $\frac{d}{du} \tanh u = \operatorname{sech}^2 u$,

$$\frac{d^2}{du^2} \operatorname{sech} u = -\frac{d}{du}(\operatorname{sech} u \tanh u) = \operatorname{sech} u \tanh^2 u - \operatorname{sech} u \operatorname{sech}^2 u = \operatorname{sech} u \left(2 \tanh^2 u - 1\right).$$

Chain rule gives the claim. $\square$

Now, we justify differentiation under the integral sign for second-order derivatives.

**Lemma 14** (Differentiation under the integral sign (second order)). *For fixed $h > 0$ and all $s \in [0, 1]$,*

$$N''(s) = \int_{[0,1]} \eta_{orig}(s_{orig}) \frac{1}{Z(s_{orig}, h)} \frac{\partial^2}{\partial s^2} \operatorname{sech}\left(\frac{s - s_{orig}}{h}\right) dp(s_{orig}),$$

$$D''(s) = \int_{[0,1]} \frac{1}{Z(s_{orig}, h)} \frac{\partial^2}{\partial s^2} \operatorname{sech}\left(\frac{s - s_{orig}}{h}\right) dp(s_{orig}).$$

*Proof.* Similarly to the proof of lemma 11, we want to apply the dominated convergence theorem. By lemma 13, the integrands are bounded in absolute value by $\frac{1}{h^2 Z(s_{\text{orig}}, h)}$. Also, we know from lemma 7 that $\min_{s_{\text{orig}} \in [0,1]} Z(s_{\text{orig}}, h) = h \arctan\left(\sinh\left(\frac{1}{h}\right)\right) > 0$. Thus $\frac{1}{h^2 Z(s_{\text{orig}}, h)}$ is an integrable dominator, so the dominated convergence theorem applies (theorem 11.32 from (Rudin, 1976)), allowing differentiation under the integral. $\square$

Similarly to what we shown for the first derivative, we show that the second derivative can be represented as a covariance under $w_s$ in the following lemma.

**Lemma 15** (Second derivative in covariance form). *Let $T(S) = \tanh\left(\frac{S-s}{h}\right)$ and expectation/covariance under $S \sim w_s$, where $w_s$ is defined as in Eq. 31. We have that:*

$$\eta''(s) = \frac{1}{h^2} \left( \operatorname{Cov}_{w_s}\left(\eta_{orig}(S), 2T(S)^2 - 1\right) - 2 \, \mathbb{E}_{w_s}[T(S)] \, \operatorname{Cov}_{w_s}\left(\eta_{orig}(S), T(S)\right) \right).$$

*Proof.* First, notice that, since $D(s) > 0$ (lemma 9), all ratios are well-defined. By lemma 14 and lemma 13, using that $T(S)^2 = \tanh^2\left(\frac{S - s_{\text{orig}}}{h}\right) = \tanh^2\left(\frac{s_{\text{orig}} - S}{h}\right)$,

$$\frac{N''(s)}{D(s)} = \frac{1}{h^2} \, \mathbb{E}_{w_s}\left[\eta_{\text{orig}}(S) \left(2T(S)^2 - 1\right)\right], \qquad \frac{D''(s)}{D(s)} = \frac{1}{h^2} \, \mathbb{E}_{w_s}\left[2T(S)^2 - 1\right].$$

From lemma 11 and lemma 10,

$$\frac{N'(s)}{D(s)} = \frac{1}{h} \, \mathbb{E}_{w_s}[\eta_{\text{orig}}(S)T(S)], \qquad \frac{D'(s)}{D(s)} = \frac{1}{h} \, \mathbb{E}_{w_s}[T(S)], \qquad \eta(s) = \frac{N(s)}{D(s)} = \mathbb{E}_{w_s}[\eta_{\text{orig}}(S)].$$

We now use the quotient rule for the second derivative:

$$\eta'' = \frac{N''D - ND''}{D^2} - 2 \frac{(N'D - ND')D'}{D^3} = \left(\frac{N''}{D} - \eta \frac{D''}{D}\right) - 2\left(\frac{N'}{D} - \eta \frac{D'}{D}\right)\frac{D'}{D}.$$

Substituting the four identities above yields

$$\eta''(s) = \frac{1}{h^2}\Big(\mathbb{E}_{w_s}[\eta_{\text{orig}}(S)(2T(S)^2-1)] - \mathbb{E}_{w_s}[\eta_{\text{orig}}(S)]\ \mathbb{E}_{w_s}[2T(S)^2-1]\Big)$$
$$- \frac{2}{h^2}\Big(\mathbb{E}_{w_s}[\eta_{\text{orig}}(S)T(S)] - \mathbb{E}_{w_s}[\eta_{\text{orig}}(S)]\ \mathbb{E}_{w_s}[T(S)]\Big)\mathbb{E}_{w_s}[T(S)].$$

Recognizing covariances,

$$\mathbb{E}[\eta(2T(S)^2-1)] - \mathbb{E}[\eta_{\text{orig}}(S)]\,\mathbb{E}[2T(S)^2-1] = \text{Cov}(\eta_{\text{orig}}(S),\, 2T(S)^2 - 1),$$
$$\mathbb{E}[\eta_{\text{orig}}(S)T(S)] - \mathbb{E}[\eta_{\text{orig}}(S)]\,\mathbb{E}[T(S)] = \text{Cov}(\eta_{\text{orig}}(S),\, T(S)),$$

we obtain

$$\eta''(s) = \frac{1}{h^2}\Big(\text{Cov}_{w_s}\big(\eta_{\text{orig}}(S),\, 2T^2(S) - 1\big) - 2\,\mathbb{E}_{w_s}[T(S)]\ \text{Cov}_{w_s}\big(\eta_{\text{orig}}(S),\, T(S)\big)\Big).$$

$\square$

**Corollary 4** (Uniform second-derivative bound). *For the* sech *kernel and any bandwidth* $h > 0$,

$$\sup_{s\in[0,1]} |\eta''(s)| \ \leq\ \frac{3}{2}\,\frac{\tanh^2(1/h)}{h^2} \ \leq\ \frac{3}{2}\,\frac{1}{h^2}.$$

*Proof.* Let $\tau := \tanh(1/h)$. Then $T(S) := \tanh\left(\frac{S-s}{h}\right) \in [-\tau, \tau]$. Hence, by Popoviciu's inequality,

$$\mathbb{V}_{w_s}(T(S)) \leq \frac{(2\tau)^2}{4} = \tau^2, \quad 2T(S)^2 - 1 \in [-1, 2\tau^2 - 1] \ \Rightarrow\ \mathbb{V}_{w_s}(2T(S)^2 - 1) \leq \frac{(2\tau^2)^2}{4} = \tau^4.$$

Since $\eta_{\text{orig}}(S) \in [0,1]$, $\mathbb{V}(\eta_{\text{orig}}(S)) \leq 1/4$. Using Cauchy–Schwarz,

$$|\text{Cov}(\eta, 2T^2 - 1)| \leq \sqrt{\mathbb{V}(\eta)\,\mathbb{V}(2T^2 - 1)} \leq \tfrac{1}{2}\tau^2, \qquad |\text{Cov}(\eta, T)| \leq \sqrt{\mathbb{V}(\eta)\,\mathbb{V}(T)} \leq \tfrac{1}{2}\tau,$$

and $|\mathbb{E}[T]| \leq \tau$. Therefore, lemma 15 gives

$$|\eta''(s)| \ \leq\ \frac{1}{h^2}\Big(\tfrac{1}{2}\tau^2 + 2\tau \cdot \tfrac{1}{2}\tau\Big) \ =\ \frac{3}{2}\,\frac{\tau^2}{h^2} \ \leq\ \frac{3}{2}\,\frac{1}{h^2}.$$

$\square$

## D.2 WHY NOT A TRUNCATED GAUSSIAN?

Another plausible choice for the kernel would have been the (truncated) Gaussian. Let $\phi(t) = \frac{1}{\sqrt{2\pi}}e^{-t^2/2}$ and $\Phi(t) = \int_{-\infty}^{t} \phi(u)\,du$ be the standard normal pdf and cdf. Fix a bandwidth $h > 0$. For each $s_{\text{orig}} \in [0,1]$ define the *truncated Gaussian* density on $[0,1]$ centered at $s_{\text{orig}}$,

$$Z(s_{\text{orig}}, h) \ = \ \Phi\!\left(\frac{1 - s_{\text{orig}}}{h}\right) - \Phi\!\left(\frac{-s_{\text{orig}}}{h}\right) \ \in\ (0, 1), \tag{32}$$

$$k(s \mid s_{\text{orig}}) \ = \ \frac{1}{h\,Z(s_{\text{orig}}, h)}\,\phi\!\left(\frac{s - s_{\text{orig}}}{h}\right). \tag{33}$$

However, in this paper we decided not to use such a kernel because of the rate of its derivative. In particular, we have that:

$$\frac{\partial}{\partial s}k(s \mid s_{\text{orig}}) \ = \ -\frac{s - s_{\text{orig}}}{h^2}\,k(s \mid s_{\text{orig}}) = \mathcal{O}\!\left(\frac{1}{h^2}\right).$$

By following a similar reasoning to the one done for the sech kernel (using the dominated convergence theorem, representing the derivative of $\eta$ as a covariance, and bounding the covariance), we would have got a bound in the order of $\mathcal{O}\left(\frac{1}{h^2}\right)$ for the first derivative, which is worse than the one we obtained for the sech kernel ($\mathcal{O}\left(\frac{1}{h}\right)$). For this theoretical reason, we opted for the more suitable sech kernel.

## E    IMPLEMENTATION DETAILS

**Plug-in bandwidth.**    We pick $h'$ (the bandwidth of $k'$) by minimizing a one-dimensional proxy built from the $h'$-dependent terms in the smoothing-error bound (Lemma 6): the bias terms are $b_1 \sum_{i \in T} w_i(s') |s' - s_i|$ and $\frac{b_2}{2} \sum_{i \in T} w_i(s') (s' - s_i)^2$, and the stochastic term is $\frac{1}{2} \sqrt{\sum_{i \in T} w_i(s')^2}$. Approximating these sums by their continuous Epanechnikov-kernel moments for locally uniform score density (ignoring boundary effects), with $K(u) = \frac{3}{4}(1 - u^2) \mathbf{1}\{|u| \leq 1\}$ and $u = (s' - s_i)/h'$, yields $\sum_i w_i |s' - s_i| \approx h' \int |u| K(u)\, du = \frac{3}{8} h'$ and $\sum_i w_i(s' - s_i)^2 \approx h'^2 \int u^2 K(u)\, du = \frac{1}{5} h'^2$, hence $a = \frac{3}{8} b_1$ and $b = \frac{1}{10} b_2$ in the proxy

$$f(h') = a\, h' + b\, h'^2 + \frac{c}{\sqrt{h'}}.$$

For the stochastic term we use the standard approximation $\sqrt{\sum_i w_i^2} \approx \sqrt{\|K\|_2^2 / (|T| h')}$ (with $\|K\|_2^2 = \int K(u)^2\, du = 3/5$ for Epanechnikov) and take a mildly conservative constant in implementation, giving $c = \frac{1.15}{2\sqrt{2|T|}}$. Setting $f'(h') = 0$ gives $a + 2bh' - \frac{1}{2} c\, h'^{-3/2} = 0$; multiplying by $h'^{3/2}$ and substituting $t = \sqrt{h'}$ yields the quintic $2b\, t^5 + a\, t^3 - \frac{1}{2} c = 0$, and we recover the bandwidth as $h' = t^2$. We solve by a few Newton steps and clip to $[10^{-4}, 1/4]$. This choice is an implementation detail; the theory only requires convex weights summing to one.

**Validation subsampling**    For speed we evaluate the held-out quantities on a uniformly sampled subset $V_k' \subseteq V_k$ per fold, with size $\min\{15000, \max(5000, 0.05\, n)\}$. Each validation point is still scored by a model trained without it, so the conditional independence and i.i.d. assumptions required by the empirical-Bernstein step remain valid; subsampling only increases variance (slightly loosening the bound) and does not alter $\delta$–allocation or correctness.

**Shift aggregation in bucketing.**    Here, "bucketing" means uniform bins in score space: for a given bucket count $B$ we use an (approximately) equal-width grid over $[0, 1]$ with bin width $\approx 1/B$, and the "shift" moves the internal bin boundaries by a small fraction of one bin width (while keeping the endpoints at 0 and 1). This is not quantile/equal-count binning. We then evaluate the Lipschitz+bucketing bound for every candidate partition defined by a bucket count $B$ and a fractional shift $r \in \{0, \ldots, R-1\}$ (offset $r/B$); using a per-candidate confidence level $\delta/(BR)$, we take the *minimum* bound over all $(B, r)$. By a union bound, this aggregate is valid at overall level $\delta$.

## F    IMPLEMENTATION OF THE PERTURBATION

One advantage of our perturbation method is that, unlike bucketing, it is compatible with training by back-prop. This means we can perturb during classifier training, increasing resilience to perturbation at test time. To do that we change the standard cross entropy loss to.

$$L = -\frac{1}{N} \left[ \sum_{i=1}^{N} y_i \mathbb{E} \log s_{1,i} + (1 - y_i) \mathbb{E} \log s_{0,i} \right] + L_m, \tag{34}$$

Where $y_i$ are the ground-truth binary labels, $\mathbb{E} \log s_{1,i}$ and $\mathbb{E} \log s_{0,i}$ are the expectation of the two log perturbed probabilities, computed using Gauss Legendre quadrature in practice.

The loss $L_m$ directly compares the logits of pairs of samples across classes, ensuring class separation. Specifically, for each example $i$ in a batch, we first consider the logit difference

$$z_i = \hat{l}_1 - \hat{l}_0,$$

where $\hat{l}_1$ and $\hat{l}_0$ denote the logits for the positive and negative classes, respectively. We now restrict attention to near pairs $(i, j)$, defined by the condition $|s^i - s^j| < 2h$ i.e. where the probabilities are close. For each such pair, the margin gap is given by

$$g_{ij} = z^i - z^j.$$

The loss penalizes cases where this margin is smaller than the target $2h$, using a smooth softplus penalty. Formally, the loss is defined as

$$L_m = \frac{1}{|C|} \sum_{(i,j) \in \mathcal{N}} \log\big(1 + \exp\big(2h - g_{ij}\big)\big),$$

where $C$ is the set of near positive–negative pairs. This formulation encourages the logit difference of positive samples relative to negatives to be at least $2h$, thereby pushing the classifier towards maintaining a larger decision margin.

## G  DATASETS

We perform perturbation experiments on three binary classification tasks:

- **IMDb:** A text data set of movie reviews with associated binary labels for sentiment (positive and negative) (Maas et al., 2011). For this task we fined-tuned a pre-trained BERT back-bone with a sequence classification head. We used $1,000$ training examples, $250$ validation examples and $500$ test examples to compute reported metrics.

- **Spam Detection:** A text dataset of emails labeled as either 'spam' or 'not-spam' Schulte et al. (2024). As for the IMDb data set, we fined-tuned a pre-trained BERT back-bone with a sequence classification head, using $1,000$ training examples, $250$ validation examples and $500$ test examples.

- **CIFAR-10:** An image data set containing 10 distinct classes: airplane, automobile, bird, cat, deer, dog, frog, horse, ship and truck (Krizhevsky, 2009). We make a binary task on this dataset by grouping means of transport (airplane, automobile, ship, truck) and animals (bird, cat, deer, dog, frog, horse). We fine-tune a pre-trained visual transformer (ViT) with a classification head. We use 2500 training examples, 250 validation examples and 500 test examples.

- **Amazon Polarity:** A large-scale text dataset of Amazon product reviews labeled with binary sentiment (positive and negative) (McAuley & Leskovec, 2013).

- **Phishing:** A dataset of websites' URLs with binary phishing and not-phishing labels (Li, 2024). We fine-tuned a BERT-based classifier model with 10000 training examples and 250 validation examples. We then use 500000 examples as test.

- **Civil Comments:** A large data set of text comments from an archive of the Civil Comments platform; a commenting plugin for independent news sites (Borkan et al., 2019). In the original set, the comments have a series of scores assigned to them, delving into different aspects of harmfulness and toxicity. We make a binary data set based on the parent toxicity score; if this score is greater than 0, the label is 1 (toxicity present) and 0 otherwise. As for the phishing data set, we use $10,000$ training examples and $250$ validation examples to fine-tune a BERT-based text classifier. We use on 3.8M as test set.

- **Yelp Polarity:** A data set of Yelp reviews, where the star ratings have been binarised ($\geq 3$ stars is considered as positive) (Zhang et al., 2015). We fine-tuned a BERT-based classifier model with 10000 training examples and 250 validation examples. We then use 560000 examples as test.

## H  DETAILS ON THE APPROXIMATION OF THE CALIBRATION FUNCTION

Figure 5 shows the reconstruction of the true calibration function $\eta$ (black) obtained using the NW (blue) and TV (red) techniques, for various sizes of the dataset. Note that increasing the dataset size increases reconstruction fidelity. Also note that the TV denoiser is regularized towards a constant function, meaning it achieves limited fidelity near the boundary regions and the bottom of the parabola.[9]

---

[9]Note that the effect diminishes with increased dataset size, as can be expected given the estimator is consistent.

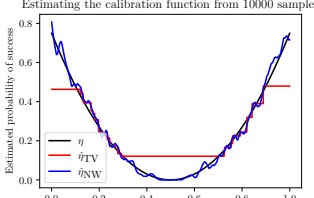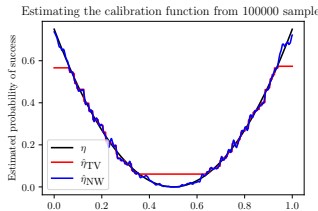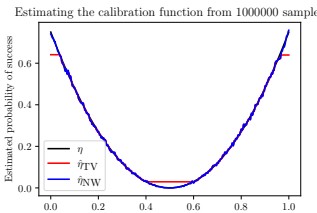

Figure 5: Reconstruction of the calibration error for various dataset sizes.

# I LLM USAGE

In accordance with the ICLR LLM usage policy, we disclose that large language models (LLMs) were used as tools during this project. First, LLMs were employed to draft concise *code prototypes from mathematical descriptions* (e.g., turning a specification of the bounds into runnable Python). All such code was subsequently scrutinized by the authors; the final implementations and experimental results are the product of human verification. Second, LLMs were used in an *iterative ideation loop for conceptualizing proofs*. The final version of formal statements, proofs, as well as the entirety of the paper's exposition were written by humans. Importantly, the paper's *central idea*—that injecting noise yields a *smooth* calibration function enabling tighter bounds—was conceived by a human author; LLMs did not originate this insight. LLMs were also used to polish the writing and as tools to find relevant related work.

