# OpenReview forum: "Measuring Uncertainty Calibration"
_ICLR.cc/2026/Conference — ICLR 2026 Poster_

### Official Review · Reviewer_2RhD · 2025-10-31

**Soundness:** 3
**Presentation:** 2
**Contribution:** 2
**Rating:** 2
**Confidence:** 4

**Summary:**

The paper addresses the problem of estimating the L1 calibration error of a binary classifier from a finite dataset. The paper has two main contributions:
(1) upper bounds on the calibration error under bounded variation assumptions using a variant of total variation denoising, and
(2) a perturbation method that enforces bounded derivatives, enabling kernel-based estimation. Both approaches are non-asymptotic and distribution-free.

**Strengths:**

- The proposed methods are non-asymptotic and distribution-free
- The empirical validation includes both synthetic experiments with known ground truth and real-data experiments. Real-data experiments (Amazon Polarity, CIFAR, IMDB, Spam) demonstrate practical applicability.

**Weaknesses:**

**Missing related work**

The paper would benefit from a more complete discussion of recent literature, e.g.
https://proceedings.neurips.cc/paper_files/paper/2020/file/26d88423fc6da243ffddf161ca712757-Paper.pdf also addresses distribution-free calibration in binary classification, establishing fundamental limits and impossibility results in the absence of distributional assumptions.

**Insufficient experimental analysis**

The experimental section is almost entirely descriptive rather than analytical. It reads more like preliminary notes or a work-in-progress draft than a finished scientific contribution.
For example, the paragraph on Real Data Experiment provides only vague descriptive claims without referencing any table/figure, or any scientific insight into why the method performs better.
The lack of analysis, missing experimental details, and superficial treatment of results suggest rushed submission.

**Overall presentation of the paper can be improved**

The paper would benefit from clearer descriptions of tables and figures, explicitly stating what is shown and what conclusions should be drawn. For instance, the table below Figure 3 lacks a caption and is not referenced in the text.
Footnote 6 appears incomplete, as it does not contain any text.
Such inconsistencies and lack of polish suggest insufficient attention to detail in the preparation of the paper and raise concerns about whether similar oversights exist in the rest of the paper.

**Overstating contributions**

The conclusion that “we empirically demonstrated it is possible to measure calibration error on a real task of practical importance” might be overstated given that the result is an upper bound, not a point estimate. It would be fairer to emphasize certification rather than estimation per se.

**Limitations not discussed**

For example, the limitation to binary classification is mentioned only in passing, yet this significantly restricts applicability given that most real-world calibration problems are multiclass. A thorough limitations section would strengthen the paper by setting appropriate expectations.

**Questions:**

How should h be chosen in practice?

ECE uses finite binning and is computationally cheap. How far apart are the derived bounds from ECE estimates?

---

> ### Author Response · Authors · 2025-11-13
> **Review response.**
>
> We thank the reviewer for the careful feedback and appreciate the opportunity to improve the paper.
>
> Concerning weaknesses:
> 1. [related work] Thank you for pointing us to the work of Gupta et al. We weren’t aware of it at the time of submission and will certainly discuss it in the related work section. We think the work of Gupta et al. is complementary to ours. Indeed, Gupta et al. (2020) studied whether calibration guarantees are theoretically possible. On the other hand, our work shows how to measure calibration reliably in finite data, with certified, distribution-free guarantees and minimal assumptions. Also note that while Gupta et al (rightly) say one needs assumptions to get a handle on calibration, they emphasize binning. This is very different from our approach where perturbation leads to finite-derivative guarantees on the calibration function, which allows sample-efficient estimation. If you think we missed other important papers, please let us know which ones - we are very keen to improve our paper.
> 2. [experiments] The real data experiment results are in Figure 3. Sorry for not referencing the Figure in the text - we will fix that in the updated version of the paper.  Concerning why each method performs better: NW performs better than TV because it has a better rate (this is a theoretical insight); NW performs better than Lipschitz bucketing because it has better constants (an empirical insight). We will make this more clear in an updated version of the paper. We will also add more experiments with real data to the updated paper. Also, if you are aware of other relevant classification datasets, please do suggest them and we will make this section larger.
> 3. [presentation] The table below Figure 3 was intended to be in the text inline but we acknowledge the presentation of this particular table could be improved with a table caption. We will do that in an updated version of the paper. The empty footnote 6 will be removed. We will also carefully proofread the paper for other cases of bad presentation.
> 4. [contributions] We will rephrase the statement of contributions in the conclusion, emphasizing the fact that our upper bound comes with a correctness certificate like you suggest. Note that, while we don’t do that in the paper, our techniques also work for lower bounding the calibration error (the change is straightforward). Having a simultaneous high-probability lower and upper bounds counts as “estimating” a random quantity. This reinforces that our approach constitutes estimation with guarantees rather than mere bounding.
> 5. [limitations] We will provide a more extensive discussion of limitations in an updated version of the paper. Concerning the fact that we do binary classification (rather than multiclass), we agree that the multiclass is an interesting avenue for further work, but at the same time we think we have enough interesting contributions about the binary case to warrant publication at this stage.
>
> Answers to questions:
> 1. [choosing h] One way to choose h is to measure AUROC for different values of h (similar to our Figure 1) and choose the largest h that doesn’t cause performance degradation (in AUROC). We will include this insight in the updated paper.
> 2. [comparison with ECE] We will compare our upper bounds with ECE values in an updated version of the paper. We did not initially opt to make this comparison because ECE is a heuristic (unlike our principled bounds).
>
> References:
> 1. Gupta, Chirag, Aleksandr Podkopaev, and Aaditya Ramdas. "Distribution-free binary classification: prediction sets, confidence intervals and calibration." Advances in Neural Information Processing Systems 33 (2020): 3711-3723.

---

> ### Author Response · Authors · 2025-11-25
> **Review Response After Revision**
>
> Thanks again for your review of the paper. We have now uploaded a new revision.
>
> 1. We discussed Gupta et al. in the related work section of the revised paper.
> 2. We improved the presentation of the whole paper and especially the experimental section, fixing missing references. We added three new datasets for the real data experiment. We also added an analysis of the ECE heuristic (it is extremely competitive for the first three choices of synthetic experiments, but fails completely for the fourth).
> 3. We have proofread the paper again, improving presentation. We also added a figure on page 2, improving presentation.
> 4. We have rephrased the contributions along the lines you suggested.
> 5. We have provided a limitations section at the end of the main part of the paper.
>
> We hope these changes address your concerns. Please do reach out if you have other suggestions for improving our work.

---

> ### Author Response · Authors · 2025-11-28
> **Please reply to author response.**
>
> Hi reviewer 2RhD,
>
> If you can, would it be possible for you to have a look at our author response and our revised version of the paper?
>
> Thanks,
>
> authors

---

### Official Review · Reviewer_UCZs · 2025-11-01

**Soundness:** 3
**Presentation:** 2
**Contribution:** 3
**Rating:** 6
**Confidence:** 3

**Summary:**

Paper introduces two new approaches for estimating binary classifier output calibration and especially upper bounds for calibration error (CE). These are  derived under bounded variation (with total variation denoising), and perturbation of classifier outputs (with kernel density estimator). Empirical evaluation shows the performance under perturbations (real data), and comparison of proposed bounds against Lipschitz bucketing on gap between the upper bound and true CE (synthetic data) and upper bound tightness (real data) with promising results.

**Strengths:**

The proposed methodologies are generally sound. To my knowledge, the derived estimators are new contributions to binary classification calibration domain. Paper is well-written and new calibration error upper bounds are motivated by the limitation of previous works and literature. Proposed theoretical results are supported by experimental results on both synthetic and real data. Although targeted on very specific classification setup, the contribution could provide useful information for the community, especially for those working on trustworthy ML. Overall, well-defined contributions to specific problem.

Summary of strengths
- Theoretically sound
- Motivated by the limitation of previous work
- Novel contributions (theoretical with techniques that could apply in practice)

**Weaknesses:**

There are also some weaknesses affecting the clarity of the presentation and significance of the results. First, although the motivation and derivation of the proposed theoretical results seem ok, the presentation could be improved and supported by illustration of the problem at the beginning, including the problem setup and possible limitations of previous work. Second, the experimental evaluation could be more versatile to fully support the proposed techniques, including the comparison with several real datasets (in addition to Figure 3), and other previous approaches presented in the related work section (in addition to Lip+Bkt). Also, different metrics of calibration accuracy and computational efficiency, and statistical significance of the results in comparison, could improve the practicality and the significance of the proposed contributions  and claims.

Summary of weaknesses
- Lack of illustrative example at the beginning  (motivation of the calibration problem setup, and limitation of previous approaches of vanilla binning etc.)
- Limitations in experiments (comparison to other previous approaches, performance metrics, computational complexity)

**Questions:**

- Why upper bounds on real data (Figure 3) is showed only for Amazon polarity set and not for other datasets?
- Would it be possible to show the performance also against other baselines mentioned in the related works? and with other performance metrics related to calibration accuracy?
- What is the computational complexity / efficiency of the proposed approaches?
- Are the results statistically significant?

Other minor comments:
- expectaiton -> expectation (page 3)
- Table X (after Figure 3) does not have caption is not referred in the text at all.

---

> ### Author Response · Authors · 2025-11-13
> **Review response.**
>
> Thanks for the review!
>
> Concerning weaknesses:
> 1. [presentation] We will improve the presentation in an updated version of the work. Specifically, we will provide an illustrative example in the beginning, simplify long sentences, improve figure readability and referencing and provide clearer takeaways in section endings.
> 2. [experiments] We will add more real-world datasets to the empirical comparison in an updated version of the paper. We will also compare with ECE (stressing it is a heuristic that doesn’t come with guarantees, unlike our approach). We will also add a discussion of other ways of measuring calibration to the paper (L2-calibration error and expected KL as arising from the Murphy decomposition), as well as a measurement of computational efficiency. Concerning statistical significance, our results are significant (see also answer to your question 4).
>
>
> Concerning your questions:
> 1. We will include the upper bounds for other datasets in a revised version of the paper.
> 2. The main missing baseline is ECE. We didn’t initially include it because it is a heuristic (unlike our principled method). We will include it in the updated version of the paper. Also, we would be very grateful if you could suggest other baselines to compare to (we aren’t aware of any other principled approaches to measuring L1 calibration error in our setting, but maybe we have missed something). Concerning other metrics (alternatives to L1 Calibration error), we will discuss them in the updated paper.
> 3. We will report on the computational efficiency in the updated version of the paper. We were able to process datasets of sizes up to 10^7 many times in a matter of minutes, so computational cost is unlikely to be a practical bottleneck. Note that, while a naive implementation of the kernel method we use is quadratic in sample size, in practice our implementation is linear since we exploit the fact that the kernel between far points is effectively zero, allowing us to implement a windowing scheme. We will discuss this more in an updated version of the paper.
> 4. The results are statistically significant. In fact, in figure 2, the confidence bars were so small as to be invisible in the plot (we will make this clear in the updated paper).

---

> ### Author Response · Authors · 2025-11-25
> **Review Response After Revision**
>
> Thanks again for your review of the paper. We have now uploaded a new revision.
> 1. We have made changes to the way the paper is presented. Specifically, we have included a new introductory figure on the second page, simplified the phrasing, improved referencing and rewritten the experimental section to both improve presentation of the takeaways and include the new results (comparison to the ECE heuristic). We also include an updated description of related work.
> 2. Concerning the experiments, we have added three additional real datasets to the paper (see figure 4). We also compared to the ECE heuristic experimentally (see figure 3). The ECE heuristic is extremely competitive for the first three choices of synthetic experiments, but fails completely for the fourth. We added an appendix discussing other alternative metrics of calibration accuracy (appendix A). We also added a discussion of computational efficiency (it is log-linear at worst for all methods and very fast in practice) and statistical significance (results are significant).
>
> In addition, we made many changes to the overall presentation of the paper.
> We hope these changes address your remaining concerns. Please do let us know if you have other suggestions for improving our work.

---

### Official Review · Reviewer_6rA7 · 2025-11-02

**Soundness:** 4
**Presentation:** 4
**Contribution:** 4
**Rating:** 8
**Confidence:** 3

**Summary:**

The paper proposes two variants of upper-bounded calibration error evaluation methods: bounded variation and bounded derivatives.
Their method of bounded derivatives achieves the tightest bound compared to the baseline.

**Strengths:**

* The paper introduced novel ways to calculate bounds for calibration error. The authors explained these bounds in detail and provided proofs. The explanations in the main part of the paper were very well written, providing most of the necessary intuitions and insights to be able to follow the paper.
* The related work is well-connected to this work.
* The experiments complement the paper and showcase that their method is giving the best results.

**Weaknesses:**

* The paper would have benefitted from illustrations about the difference of $\eta$ and $\hat{\eta}$ for TV denoising and for kernel smoothing. Such illustrations could be done for some moderately non-monotonic function, e.g. with total variation above 1 but below 2. While the descriptions were all understandable for me eventually, I think most readers would benefit from such illustrative figures. There was some space remaining for this (if I understand correctly that LLM usage, reproducibility statement, and ethics statement can be outside the 9-page-limit).

* Some of the notation was not described in equations (4) and (5). In particular, TVB(delta') in Eq.(4) and B(delta) in Eq.(5) have not been explicitly defined. I did not find the explicit definitions in the Appendix either. However, conceptually it is understandable what these terms mean.

* It would also be nice to see experiments about how classical calibration evaluation methods such as ECE perform in relation to the proposed methods. It is understandable that ECE does not provide such useful upper bounds, but a comparison would help to understand better the shortcomings of ECE.

* It was not clearly justified why Lipschitz bucketing was used in the experiments as the only earlier method to compare with.

* Minor issue: at the end of section 3, in the paragraph about Concentration, the notation starts with $\hat{\sigma}^2_n$ but then switches to $\hat{\sigma}^2_{X_i}$. The latter is confusing, because it does not depend only on a particular $X_i$, but all $X_1,\dots,X_n$. Hence, I suggest staying with $\hat{\sigma}^2_n$.

* The section on related work seems to be a bit out-of-date, with one reference in 2024, but the rest are older. Several works on evaluating classifier calibration published in 2025 have been omitted:

Dieye et al (2025). When standard calibration metrics fail in evaluating classifier calibration: A simulation study. In ClaDAG 2025.

Kängsepp et al (2025). On the usefulness of the fit-on-test view on evaluating calibration of classifiers. Machine Learning, 114(4), p.105.

Maalej et al (2025). Evaluating Calibration Techniques for Reliable Predictions. In International Conference on Machine Learning and Soft Computing (pp. 159-175)

**Questions:**

1) What are the exact definitions of TVB(delta') in equation (4) and B(delta) in equation (5)?

2) Why was Lipschitz bucketing used in the experiments as the only earlier method to compare with?
	* Are there any other methods for comparison, or is Lip+Bkt the only or the best bounded method?
	* In row 420, the phrase "our proposed" goes with NW but not with TV denoising. This is slightly confusing. I thought TV denoising was also first used in the context of calibration in this paper. Can the TV denoising method be also stated as "our proposed"?

3) How do classical calibration evaluation methods such as ECE perform in relation to the proposed methods? It is understandable that ECE does not provide such useful upper bounds, but a comparison would help to understand better the shortcomings of ECE.

4) One usecase of measuring calibration is when comparing different calibration methods and deciding which method works best, by giving the smallest calibration errors. Could the proposed methods be used to rank calibration methods? Would calculating the lower bound in addition to the upper bound be helpful with ranking which calibration method is better?

5) Why is Lip+Bkt not given as an option in practical advice, as it has tighter bounds than TV?

---

> ### Author Response · Authors · 2025-11-13
> **Review response.**
>
> Thanks for the review!
>
> Concerning weaknesses:
> 1. We will provide illustrations for the reconstructed vs true eta in an updated version of the paper.
> 2. Thanks for catching these! We will improve the presentation in an updated version of the paper. TVB is the right-hand-side of Corollary 1 (but isn’t currently denoted as such). We will also remove the confusing notation B(delta) – it should read TVB(delta).
> 3. We will provide an empirical comparison with ECE in an updated version of the paper.
> 4. Other than Lipschitz bucketing, it was hard for us to find methods with similar guarantees that we could compare to. Perhaps we have missed something – do feel free to make suggestions.
> 5. We will fix the notation used for the variance estimate in Bernstein concentration.
> 6. We will compare our work to the papers you suggest in the related work section of the updated paper.
>
> Concerning questions:
> 1. They both mean the same thing (we accidentally used two shortcut notations for the same concept). TVB is the right-hand-side of Corollary 1. We agree it is confusing and will fix this.
> 2. See response to weakness 4. Concerning row 420, you are right that we propose both NW and TV denoising for the calibration problem. We will fix the way this is presented.
> 3. We will include an empirical comparison with ECE in a revised version of the paper.
> 4. Our method can be used to rank calibration methods although we would prefer to leave that to further work. A simultaneous lower bound is achievable in a straightforward way using our theoretical technique.
> 5. Lip+Bkt does give tighter bounds compared to TV and can be useful. However, the problem is with justifying the Lipschitz assumption. If we don’t know a priori that the calibration function arising from our classifier is Lipschitz (as is most often the case), we cannot use the method directly and have to perturb the scores. If we do perturb the scores, we are better off with NW. TV avoids this issue because it does not require the calibration function to be Lipschitz, only of bounded variation (a much more benign assumption).

---

> ### Author Response · Authors · 2025-11-25
> **Review Response After Revision**
>
> Thanks again for your review of the paper. We have now uploaded a new revision.
> 1. We have provided illustrations showing the difference in the true and estimated calibration function (See appendix H and the new figure 5).
> 2. We fixed presentation issues and clearly defined TVB.
> 3. We compared to the ECE heuristic experimentally (see figure 3). The ECE heuristic is extremely competitive for the first three choices of synthetic experiments, but fails completely for the fourth.
> 4. We did not include baselines other than Lipschitz bucketing or ECE in the revision, because we are not aware of any others that match our setting.
> 5. We fixed the presentation of Bernstein concentration.
> 6. We have expanded the related work section.
>
> In addition, we made many changes to the overall presentation of the paper.
> We hope these changes address your remaining concerns.

---

### Author Response · Authors · 2025-11-30
**Summary comment for the new area chair.**

### Our Motivation for Posting
After the paper discussion phase got interrupted, we wanted to take this opportunity to summarise our rebuttal. We had also posted replies to each review before the area chair reassignment, addressing each reported weakness separately, but we didn't manage to get any replies from the reviewers.

### Summary of Weak Points As Identified by Reviewers
The most critical review we got was 2RhD. The most important problems the review has identified were (1) a missing paper in the related work section (2) a suggestion to add more experiments (3) a suggestion to improve presentation. We have submitted a revised version of the paper that addresses these points. Specifically, we have updated the related work section, included new experiments on more real world datasets and compared our principled approaches to the ECE heuristic, in addition to improving the presentation of several sections of the paper. We are confident these changes address the reviewer's concerns.

Reviewer UCZs was more positive, but initially had some concerns about the scope of experiments and the presentation. We believe these were addressed in our response.

Reviewer 6rA7 was very positive about our paper - see our reply to them for a discussion of minor details.

### Our Take on the Discussion
While we got one critical review, the reasons for the low score appear to not cut very deep. It is not very fair to reject a paper because it missed one (largely orthogonal) piece of related work (we added the paper to related work during the rebuttal) and because of minor presentation issues. The real issue (more experimental evaluation and comparison with ECE)  was addressed in the revised version.

### Why we Believe the Paper Should be Accepted
Our paper fills an important need of the ICLR community, proposing certified bounds for the calibration error of a classifier. Our two main results (Propositions 1 and 2) provide practical recipes for bounding the calibration error under different assumptions (bounded variation and bounded derivatives). In addition, we crucially introduce a perturbation technique that makes the bounded derivatives assumption hold, filling a gap in existing approaches (most of which were either totally heuristic or assumed the Lipschitz property without justification).

---

### Author Response · Authors · 2025-11-30
**Comment for the new area chair.**

Dear new area chair,

We have summarised the discussion and our thinking about the paper in the post below.

If you have any more questions / doubts about the paper, please do post them!

Thanks,

authors

---

### Meta-Review · Area_Chair_s1F1 · 2026-01-06

**Summary:**

This paper provides upper bounds on calibration for binary classification, estimated from a finite data set, under various reasonable assumptions. Overall, the results of this paper are quite novel, and the paper is well written and easy to follow. While there are minor notational issues, a few missing related works (which the authors have already addressed), and more experiments that could illustrate the proposed bounds, this paper overall provides missing analysis and theoretical grounding for an important topic in machine learning. This work is high quality and of interest to the ICLR community, and therefore, I recommend acceptance.

**Reviewer Concerns:**

Though the reviewers did not respond to the rebuttal, the authors have adequately addressed all concerns and weaknesses. Much of what the reviewers brought up was (1) small notational issues/confusions, (2) clarity and illustrative examples, (3) missing references to related work, and (4) discussion of contributions and limitations. From my reading, all of these points are relatively minor and were thoroughly addressed by the authors in their rebuttal and revision.

**Reviewer Scores:**

* Reviewer 6rA7 would likely have appreciated the clarifications and kept their score.
* Reviewer UCZs would likely have appreciated the discussion and additional experiments. I do not know whether they would have increased their score, but it would likely have remained the same.
* The low score from 2RhD is surprising given their relatively minor weaknesses. I don’t know how they would have engaged with the rebuttal, but I did not take their low score into account (though I did consider the content of their review).

---

### Decision · Program_Chairs · 2026-01-26

Accept (Poster)